# The Four Color Theorem for Cell Instance Segmentation

**Ye Zhang**[1,2] **Yu Zhou**[2] **Yifeng Wang**[3] **Jun Xiao**[1] **Ziyue Wang**[4] **Yongbing Zhang**[1] **Jianxu Chen**[2]

## Abstract

Cell instance segmentation is critical to analyzing biomedical images, yet accurately distinguishing tightly touching cells remains a persistent challenge. Existing instance segmentation frameworks, including detection-based, contour-based, and distance mapping-based approaches, have made significant progress, but balancing model performance with computational efficiency remains an open problem. In this paper, we propose a novel cell instance segmentation method inspired by the four-color theorem. By conceptualizing cells as countries and tissues as oceans, we introduce a four-color encoding scheme that ensures adjacent instances receive distinct labels. This reformulation transforms instance segmentation into a constrained semantic segmentation problem with only four predicted classes, substantially simplifying the instance differentiation process. To solve the training instability caused by the non-uniqueness of four-color encoding, we design an asymptotic training strategy and encoding transformation method. Extensive experiments on various modes demonstrate our approach achieves state-of-the-art performance. The code is available at https://github.com/zhangye-zoe/FCIS.

## 1. Introduction

Cell-level analysis tasks hold broad application prospects in the biomedical field. Accurate cell segmentation (Petukhov et al., 2022; Zhang et al., 2025a; Chen et al., 2024) not only provides a necessary foundation for downstream tasks such as cell counting (Falk et al., 2019), cell classification

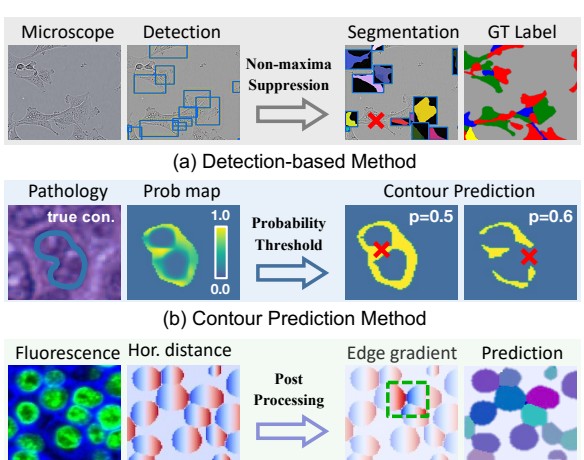

Figure 1. Existing cell instance segmentation frameworks. (a) represents the detection-based method, which cannot tackle elongated cells. (b) represents the contour prediction methods influenced by threshold value choice. (c) represents the distance mapping methods, which contains multiple tasks and relies on a post-processing process. The Red "×" indicates the segmentation mistakes.

(Cords et al., 2023; Zhang et al., 2025b), and cell tracking (Merryweather et al., 2021), but also underpins critical applications in clinical diagnostics, such as immune microenvironment analysis (Barkley et al., 2022; Kao et al., 2022) and biomarker discovery (Mann et al., 2021).

At present, cell instance segmentation models can be categorized into three primary approaches: *(a) detection-based methods* (Jiang et al., 2023), which rely on object detection frameworks (Ren et al., 2016) to localize and delineate individual cells; *(b) contour prediction methods* (Chen et al., 2016), which explicitly predict cell boundaries to achieve instance differentiation; and *(c) distance mapping methods* (Graham et al., 2019; He et al., 2021), which encode spatial relationships or distance information to separate adjacent cells. Despite their demonstrated success in cell instance segmentation tasks, existing methods face critical limitations due to the inherent diversity of cell morphologies and image characteristics across different scenarios, which impose stringent demands on model generalization.

For different segmentation methods, their problems include the following aspects, as shown in Figure 1. **First**, detection-based methods often struggle with complex cases such as

[1]School of Computer Science and Technology, Harbin Institute of Technology (Shenzhen), China [2]Leibniz-Institut für Analytische Wissenschaften – ISAS – e.V., Germany [3]School of Science, Harbin Institute of Technology (Shenzhen), China [4]Department of Electrical and Computer Engineering, National University of Singapore, Singapore. Correspondence to: Yongbing Zhang <ybzhang08@hit.edu.cn>, Jianxu Chen <jianxu.chen@isas.de>.

*Proceedings of the 42$^{st}$ International Conference on Machine Learning*, Vancouver, Canada. PMLR 267, 2025. Copyright 2025 by the author(s).

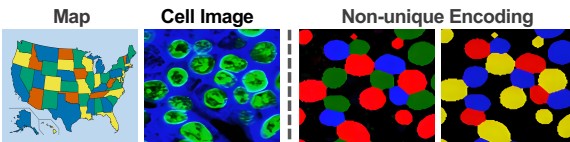

*Figure 2.* Our proposed cell encoding method is based on the four-color theorem. In the method, each cell is viewed as a "**country**," and the encoding ensures that adjacent cells have different colors.

elongated fibroblasts or overlapping cells, leading to frequently missed detections. **Second**, contour prediction methods attempt to achieve instance separation by introducing an additional contour category. However, their performance heavily depends on the contour threshold setting. **Lastly**, distance mapping methods rely on highly intricate network architectures and sophisticated post-processing workflows, which significantly increase computational overhead and model complexity. Therefore, it is imperative to develop an innovative approach that can address the shortcomings of current methods by *enhancing robustness*, *reducing computational complexity*, and *improving generalization* across diverse biomedical image modes.

The four-color theorem (Fritsch et al., 1998) offers a novel perspective on cell instance segmentation. As shown map in Figure 2, the theorem states that only four colors are sufficient to ensure that adjacent regions are assigned distinct colors (Gonthier et al., 2008). By drawing an analogy to cell images, we conceptualize each cell instance as a "**country**," while the background corresponds to the "**ocean**." This enables the development of a four-color encoding scheme that assigns unique encodings to adjacent cells. Under this framework, the instance segmentation task is reformulated as a four-class semantic segmentation problem, simplifying instance differentiation. However, the inherent non-uniqueness of four-color encodings and the class imbalance introduced by the encoding strategy pose significant challenges for model training. Directly using these encodings as supervision can lead to training instability and hinder optimization. Therefore, a well-designed training strategy is essential to address the training problem effectively.

To address the above challenges, we propose an asymptotic training architecture. Different from traditional semantic segmentation methods (Wang et al., 2018; Zhou et al., 2022), which adopt simultaneous multi-class, multi-channel output strategies, our method adopts a step-by-step approach: first distinguish foreground and background and then make category prediction within the foreground region. Our approach prioritizes high-level spatial information over fine-grained semantic details by imposing orthogonal constraints on adjacent cells, effectively solving the class imbalance problem. To mitigate the training instability caused by the non-uniqueness of the encoding, we introduce an encoding transformation method that maps the output to a minimum

color representation, ensuring consistency by removing ambiguity in encoding variations. In addition, we provide a theoretical analysis to prove the reasonability of the designs.

In summary, the contributions of this paper are five folds:

- We propose an innovative cell segmentation method based on the four-color theorem, which transforms instance segmentation into a semantic segmentation task, eliminating complex instance differentiation designs.

- We design an asymptotic training strategy incorporating a foreground prediction transformation module, greatly enhancing training stability and robustness.

- We systematically summarize the characteristics of cell distributions in medical images, demonstrating that cell coloring is inherently simpler than map coloring.

- We provide a rigorous theoretical analysis to justify the rationale and feasibility of the proposed model design.

- We validate the effectiveness of our method on three distinct types of medical image datasets. The results show that our method successfully balances performance and model complexity.

## 2. Complexity Analysis of Model Training

The advancement of deep learning revolutionizes automated cell segmentation, significantly reducing the time and effort required for manual annotation (Stringer et al., 2021; Pachitariu & Stringer, 2022; Zhang et al., 2025c). Although existing approaches show impressive performance, they are difficult to fit in various segmentation scenes and face challenges regarding training complexity and post-processing requirements. To facilitate a comprehensive comparison of existing methods, we summarize the computational complexity of the above three types of models. At the same time, the Supplementary Material provides more extensive research of related works.

**Detection-Based Methods**

The overlapping boundaries of cells remain a critical challenge in cell segmentation. Detection-based methods address this issue through a two-stage strategy: first, a detection network (Ren et al., 2016; Redmon & Farhadi, 2017) localizes cell positions; then, segmentation predictions are generated based on the detection results. Representative methods include IRNet (Zhou et al., 2020), and DoNet (Jiang et al., 2023). To enhance localization accuracy, these methods commonly incorporate non-maximum suppression (NMS) to merge highly overlapping detection boxes, thereby reducing over-prediction. However, this strategy can result in missed detections, particularly for small or irregularly

shaped cells. Additionally, detection-based approaches often exhibit high computational complexity due to the intricate detection and segmentation network designs. As shown in Table 1, these methods typically have higher parameter complexity and computational overhead regarding FLOPs.

**Contour Prediction Methods**

Contour prediction-based methods achieve instance differentiation by introducing contour semantic categories into the model's predictions. However, due to the few pixels in the contour, the segmentation for contours is often inferior to that for the background and foreground. To address this issue, two solutions are proposed. The first solution, represented by UNet (Ronneberger et al., 2015; Zhou et al., 2018), increases the loss weight of boundary to guide the model to focus more on contour. With relatively simple structural designs, these methods typically have lower parameter counts and computational complexity, as shown in Table 1. However, their performance remains suboptimal, constrained by the limited effectiveness of the loss weighting strategy. The second solution, represented by Micro-Net (Raza et al., 2019), enhances contextual perception by introducing complex network structures (Zhou et al., 2019), such as multi-scale feature fusion (Srivastava et al., 2021) and attention mechanisms (Prangemeier et al., 2020; Hörst et al., 2024). While this approach significantly improves performance compared to the former, including complex modules substantially increases model complexity, resulting in longer training times and higher computational costs.

**Distance Mapping Methods**

Distance-based cell segmentation methods, such as StarDist (Schmidt et al., 2018), CellViT (Hörst et al., 2024), and RepSNet (Xiong et al., 2025), utilize distance maps to enhance instance differentiation, especially in cases involving irregular cell shapes or densely packed regions. While these methods have shown significant success, they typically rely on multiple decoding branches that require post-processing (Graham et al., 2019; Chen et al., 2023; Meng et al., 2024) to merge the results into accurate instance segmentations. This multi-branch design increases model complexity, as the network must simultaneously handle various tasks, including distance map prediction and semantic category classification. As shown in Table 1, distance-based methods generally exhibit higher parameter complexity and computational cost than detection-based and contour prediction methods, limiting their efficiency for large-scale applications.

The four-color-theorem introduces a novel cell instance segmentation paradigm that eliminates the dedicated instance differentiation modules. This method significantly reduces training complexity by reformulating the instance segmentation task as a semantic segmentation problem. Furthermore, experimental results in Table 1 demonstrate that this ap-

| Methods | # Paras | #FLOPs | Publication |
|---|---|---|---|
| *Detection based methods* | | | |
| Mask-RCNN (He et al., 2017) | 44.66 M | 411.61 G | ICCV |
| DoNet (Jiang et al., 2023) | 67.71 | 221.64 G | CVPR |
| *Contour prediction methods* | | | |
| UNet (Ronneberger et al., 2015) | 32.14 M | 64.27 G | MICCAI |
| DCAN (Chen et al., 2016) | 41.16 M | 77.82 G | CVPR |
| CNN3 (Kumar et al., 2017) | 65.46 M | 1.06 G | TMI |
| UNet++ (Zhou et al., 2018) | 9.28 M | 35.61 G | MICCAI |
| FullNet (Qu et al., 2019) | 112.60 M | 116.92 G | MICCAI |
| Micro-Net (Raza et al., 2019) | 89.64 M | 72.96 G | MIA |
| NucleiSegNet (Lal et al., 2021) | 12.40 M | 18.19 G | CBM |
| TSFD-Net (Ilyas et al., 2022) | 21.96 M | 12.10 G | NN |
| GeNSeg-Net (Xu et al., 2024) | 87.11 M | 86.84 G | MM |
| *Distance mapping methods* | | | |
| StarDist (Schmidt et al., 2018) | 21.43 M | 92.40 G | MICCAI |
| HoverNet(Graham et al., 2019) | 49.70 M | 192.70 G | MIA |
| CDNet (He et al., 2021) | 70.55 M | 44.87 G | ICCV |
| SONNET (Doan et al., 2022) | 63.87 M | 166.75 G | JBHI |
| TransUNet (He et al., 2023) | 112.21 M | 37.67 G | MICCAI |
| CPP-Net (Chen et al., 2023) | 80.75 M | 163.10 G | TIP |
| SMILE (Pan et al., 2023) | 53.85 M | 68.58 G | MIA |
| NuSEA (Meng et al., 2024) | 55.26 M | 74.20 G | JBHI |
| CellViT (Hörst et al., 2024) | 96.81 M | 124.25 G | MIA |
| RepSNet (Xiong et al., 2025) | 28.20 M | 137.11 G | IJCV |
| *Our four-color theorem based method* | | | |
| FCIS (Ours) | 39.75 M | 58.03 G | - |

*Table 1.* The computational complexity and number of parameters comparisons. All the methods are reported for $256 \times 256$ inputs.

proach achieves substantial advantages in both parameter efficiency and computational cost compared to detection-based and distance-based segmentation methods.

## 3. Cell Encoding by Four Color Theorem

### 3.1. Greedy Algorithm for Encoding

The four-color theorem illustrates the minimum color number required to label adjacent regions without overlap, providing a novel approach to the cell instance segmentation problem. Unlike traditional instance segmentation workflows, this method transforms the instance segmentation task into a multi-class semantic segmentation problem. Based on this theory, we designed a greedy algorithm to generate four-class encoded representations for the foreground regions. The encoding process is illustrated in Algorithm 1.

We first preprocess the input image and its corresponding labels $(X, Y)$ to construct a cell graph $G = (V, E)$, where the node set $V = \{v_i \mid i = 1, \cdots, N\}$ represents the cells in the image, and the edge set $E = \{e_{i,j}\}$ represents the adjacency relationships between cells. The label $Y$ contains instance-level annotations, with each instance uniquely identified by an identification, while $e_{i,j}$ indicates that cells $v_i$ and $v_j$ are adjacent. Next, we assign color encodings to

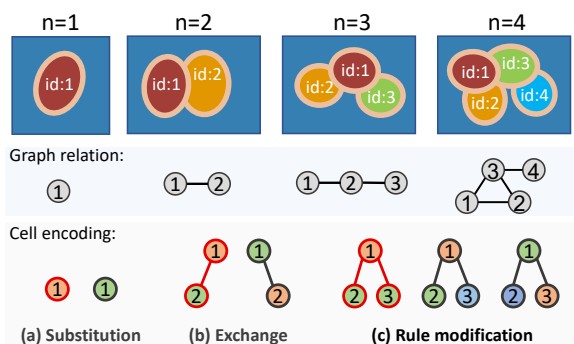

*Figure 3.* The non-uniqueness of four color encoding can be summarized as the following three cases: (a) Encoding substitution; (b) Encoding exchange, and (c) Encoding rule modification.

the nodes $v \in V$ using a greedy algorithm. Specifically, for each node $v$, we first compute its set of neighboring nodes $N(v) = \{u \mid (v, u) \in E\}$ and collect the colors $\mathcal{C}_{\text{used}}$ already assigned to these neighboring nodes as follows:

$$\mathcal{C}_{\text{used}} = \{C(u) \mid u \in N(v), C(u) \neq 0\}. \qquad (1)$$

We assign the smallest available color from the four color set $\mathcal{C} = \{1, 2, 3, 4\}$ to the current node $v$, ensuring that it does not conflict with the colors of its neighboring nodes:

$$C(v) = \min(\mathcal{C} \setminus \mathcal{C}_{\text{used}}). \qquad (2)$$

This process guarantees that two adjacent nodes $v_i$ and $v_j$ are assigned different colors, i.e., $C(v_i) \neq C(v_j)$. After encoding all cells, we generate the final segmentation mask $M$. For each pixel $p$ in the image, if the pixel belongs to a specific nucleus $v$, it is assigned the color encoding $C(v)$ corresponding to that nucleus. The final output segmentation mask $M$ provides a four class semantic segmentation representation based on the four color theorem.

### 3.2. Non-uniqueness Property of Encoding

While four-color encoding ensures heterogeneity of adjacent cell colors, its non-uniqueness may lead to convergence issues during model training. To illustrate the potential problems of this encoding more intuitively, Figure 3 presents the differences in cell encoding under various distribution scenarios. The first row shows the spatial relationships between cells, and the second row represents the corresponding cell graph structures, which cover the most common cell distribution scenarios. Combinations of these basic patterns can represent more complex cell distributions. Moreover, the third row illustrates encoding representations. In detail, the cells marked in red represent the initial encoding generated by the greedy algorithm. In contrast, the cells marked in black correspond to the equivalent encoding that satisfies the four-color theorem. The differences between the greedy algorithm's results and the equivalent encoding contain the following three cases:

**(a) Substitution**: The color of a cell is replaced with another color while maintaining the same number of colors.

**(b) Exchange**: The color assignments between two or more cells are swapped, preserving the overall color count.

**(c) Rule Modification**: Certain cells are assigned new colors, resulting in an increase in the total number of colors.

---

**Algorithm 1** Cell Encoding by Greedy Algorithm

1: **Input:** Cell graph $G = (V, E)$, where $V$ is the set of cells and $E$ represents adjacency relation.
2: **Output:** Four-color encoded mask $C(v)$.
3: Initialize color set $\mathcal{C} = \{1, 2, 3, 4\}$.
4: Initialize mask $C(v) \leftarrow 0, \forall v \in V$.
5: **for** each nucleus $v \in V$ **do**
6:     Get the set of neighbors: $N(v) = \{u \mid (v, u) \in E\}$.
7:     Collect used colors: $\mathcal{C}_{\text{used}} = \{C(u) \mid u \in N(v), C(u) \neq 0\}$.
8:     Assign the smallest available color:
9:     $C(v) \leftarrow \min(\mathcal{C} \setminus \mathcal{C}_{\text{used}})$.
10: **end for**
11: Generate segmentation mask $M$:
12: **for** each pixel $p$ in the image **do**
13:     Assign pixel $p$ to the color of the cell it belongs to:
14:     $M(p) \leftarrow C(v)$, where $v$ is the cell containing $p$.
15: **end for**
16: **Return** $M$ (Four-color annotation mask)

---

Segmentation networks typically learn semantic categories based on the object's morphological and textural features (Jain et al., 2023), while four-color encoding emphasizes the spatial relationships between cells. When faced with the issue of encoding non-uniqueness, conventional networks often struggle to converge stably (Ronneberger et al., 2015). To propose a reasonable solution, we further analyze the characteristics of the greedy algorithm in the next section.

### 3.3. Low-rank Property of Greedy Encoding

Greedy algorithms (GAs), as heuristic methods, are commonly used to generate locally optimal solutions. However, in the cell coloring problem, GAs can achieve globally optimal solutions. The reasons for this are twofold: First, the cells usually exhibit global dispersion and local aggregation, with a relatively small number of cells in each cluster, which differs from the distribution of countries. Second, adjacent cells in the image usually follow a chain-like or rectangular arrangement. Hence, each cell has much fewer neighbors. These structural properties render the cell coloring problem more straightforward than the map coloring problem.

To clarify the above fact, we statistics the number of colors distributed on each image under four-color encoding in Figure 4. The scatter plot on the left corresponds to each sample, and the box plot shows the distribution of encoding

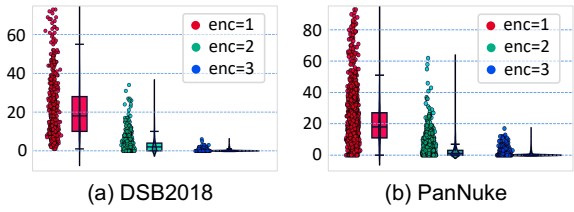

*Figure 4.* Statistics of the number of cells with different color in the DSB2018 and PanNuke datasets.

numbers. The results demonstrate that only a tiny proportion of images require more than two colors and almost no image requires four colors. Based on these, we will present the global optimal theory of greedy algorithm coloring.

**Theorem 1. Global Optimality of Greedy Coloring:** *Let $G = (V, E)$ be an undirected graph; among them, $V$ is the set of vertices, and $E$ is the set of edges. Suppose $G$ satisfies the following conditions:*

*(1) $G$ is planar, meaning it can be embedded in the plane without any edges crossing each other.*

*(2) The maximum degree of $G$, denoted $\Delta(G)$, satisfies:*

$$\Delta(G) \leq k, \quad where \quad k \leq 4. \tag{3}$$

*(3) The vertex distribution of $G$ follows a specific structure, either a chain structure (vertices are ordered linearly) or a rectangular structure (vertices are arranged in a grid pattern).*

*Then, the chromatic number with the greedy algorithm is equal to the chromatic number:*

$$\chi_{greedy}(G) = \chi(G). \tag{4}$$

*Where $\chi(G)$ denote the chromatic number, which is the minimum number of colors, and $\chi_{greedy}(G)$ denote the chromatic number obtained by applying the greedy coloring algorithm.* Some related definitions and proofs are included in the Supplementary Material.

## 4. Method Designs

### 4.1. Asymptotic Training Strategy

Previous research primarily focused on designing powerful feature extractors (Liu et al., 2022; Yu et al., 2024) or context-aware modules (Liu et al., 2021; Li et al., 2024) to enhance the network classification ability. However, in the scenario of four-color encoding, the model not only requires learning semantic features to distinguish foreground and background but also needs to learn positional information, ensuring adjacent cells are assigned distinct colors. To address the dual requirements, we propose an asymptotic training strategy as illustrated in Figure 5 (a).

### Binary Classification Semantic Prediction

Given an input image $X_i$, an encoder-decoder network is employed to generate a five-channel feature map $\hat{Y}_i \in \mathbb{R}^{H \times W \times 5}$, where $H$ and $W$ are the height and width of input. Among these channels, the first represents the background probability, and the remaining four represent the prediction of the four-color encoding. Hence, the probability map of background $\hat{Y}_b$ is extracted as follows:

$$\hat{Y}_b = \hat{Y}_i[:, 0], \tag{5}$$

where $\hat{Y}_i[:, 0]$ denotes the first channel of the prediction feature map. For obtaining the foreground probability, we use a convolution operation to transform the last four channels into a single-channel foreground probability:

$$\hat{Y}_f = \text{Conv}(\hat{Y}_i[:, 1:5]), \tag{6}$$

where $\text{Conv}(\cdot)$ represents convolutional layers. Combined the probility maps of background $\hat{Y}_b$ and foreground $\hat{Y}_f$, the binary semantic prediction can be formulated as:

$$\hat{Y}_{b,i} = \text{Concat}(\hat{Y}_b, \hat{Y}_f), \tag{7}$$

where $\text{Concat}(\cdot, \cdot)$ denotes the concatenation operation along the channel dimension. To optimize the binary semantic predictions, we define the semantic loss as:

$$\mathcal{L}_{sem} = \text{CE}(\hat{Y}_{b,i}, Y_i) + \text{Dice}(\hat{Y}_{b,i}, Y_i), \tag{8}$$

where $\text{CE}(\cdot, \cdot)$ represents the cross-entropy loss, and $\text{Dice}(\cdot, \cdot)$ is the Dice coefficient loss. Where $Y_i$ denotes the ground truth labels for binary segmentation.

### Four-Color Category Prediction

To accurately identify foreground regions and ensure distinct encodings for adjacent cells, we propose a negative sampling constraint method, as shown in Figure 5 (b). This method enforces heterogeneity for adjacent cells while preserving the accuracy of four-color encoding.

First, based on cell connectivity relationships, we sample features from the adjacent cell pairs $(v_i, v_j)$. Meanwhile, the sampled feature sets can be formulated as follows:

$$F_i = \{f_i^\alpha \mid \alpha = 1, \dots, M\}, \tag{9}$$

$$F_j = \{f_j^\beta \mid \beta = 1, \dots, N\}, \tag{10}$$

where $M$ and $N$ are the number of sampling obtained from cells $v_i$ and $v_j$, respectively, and $f_i^\alpha$ denotes the feature vector of the $\alpha$-th pixel in cell $v_i$.

To ensure that the feature representations of adjacent cells exhibit sufficient heterogeneity, we impose an orthogonality constraint in the feature space. This constraint is formulated using a cosine similarity loss:

$$\mathcal{L}_{ort} = \frac{1}{|E|} \sum_{(v_i, v_j) \in E} \text{Cos}(F_i, F_j), \tag{11}$$

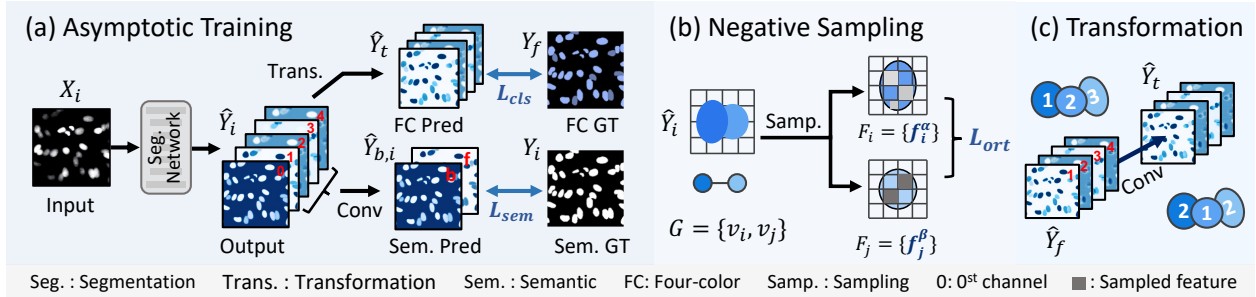

*Figure 5.* The training framework of proposed FCIS. (a) represents the asymptotic training method, (b) represents the negative sampling learning for adjacent cells, and (c) represents the encoding transformation method.

where $\text{Cos}(\cdot, \cdot)$ denotes the cosine similarity function, $E$ is the set of all edges representing adjacent cell pairs, and $|E|$ is the total number of edges. By minimizing $\mathcal{L}_{\text{ort}}$, the similarity of feature representations is suppressed, thereby enhancing the model's ability to distinguish cells.

### 4.2. Encoding Transformation

Although the orthogonality constraint ensures heterogeneous encoding, this sampling-based supervision remains weak and may not effectively guide model training. To provide stronger supervision, we introduce four-color encoding as the target label. However, the non-uniqueness of the encoding can lead to inconsistencies in supervision, potentially hindering model convergence. To mitigate this issue, we propose an encoding transformation method, whose mechanism is established in the following theorem.

**Theorem 2. Greedy Coloring Compatibility:** *In the cell instance segmentation task, let the encoding matrix generated by the greedy algorithm be:*

$$\mathbf{C} \in \mathbb{R}^{n \times k}, (k \leq 4). \tag{12}$$

*And the encoding matrix predicted by the network is:*

$$\mathbf{P} \in \mathbb{R}^{n \times k'}, \tag{13}$$

*$n$ represents the number of cells, $k$ is the number of colors used in the greedy algorithm, and $k'$ is the number of predicted encodings.*

*If the predicted encoding matrix $\mathbf{P}$ has one of the relations with the greedy encoding $\mathbf{C}$, i.e., **substitution**, **exchange**, **rule modification**. Then there exists a mapping function: $f : \mathbf{P} \to \mathbf{C}$, such that the network's predicted result can be transformed into the four-color encoding result. The detailed proof is shown in* Supplementary Material.

Based on the above theory, we propose an encoding transformation method consisting of two convolutional layers, which maps the network's predicted output $\hat{Y}_f$ into the optimal encoding $\hat{Y}_t$, as shown in Figure 5 (c). This transformation ensures adherence to the four-color encoding rules,

improving the model's overall performance and accelerating its convergence during training. Employing the transformed prediction, we compute a classification loss specific to the foreground as follows:

$$\mathcal{L}_{\text{cls}} = \text{CE}(\hat{Y}_t, Y_f) + \text{Dice}(\hat{Y}_t, Y_f), \tag{14}$$

Where $\hat{Y}_t$ and $Y_f$ represent the predicted and ground truth foreground regions, respectively. In the optimization objective, we only calculate the loss of the foreground region.

**Total Loss Function**

The overall loss function integrates the semantic, orthogonality, and classification losses and is formulated as follows:

$$\mathcal{L}_{\text{total}} = \mathcal{L}_{\text{sem}} + \lambda_1 \mathcal{L}_{\text{ort}} + \lambda_2 \mathcal{L}_{\text{cls}}. \tag{15}$$

where $\lambda_1$ and $\lambda_2$ are hyperparameters that control the importance of the orthogonality and classification losses. In the paper, we set $\lambda_1 = 2$ and $\lambda_2 = 1$. More experiment comparisons are added in Supplementary Material.

## 5. Experiments

### 5.1. Datasets

We evaluated our proposed method on multiple types of cell images, including pathological images, fluorescence-stained images, bright-field images and phase-contrast images. Specifically, the datasets used include BBBC006v1 (Ljosa et al., 2012), DSB2018 (Caicedo et al., 2019), Pan-Nuke (Gamper et al., 2020) and YeaZ (Dietler et al., 2020).

The **BBBC006v1** consists of 768 Hoechst 33342 marker-stained images, each with a resolution of $696 \times 520$ pixels. Following the dataset split used by CPP-Net (Chen et al., 2023), we divide the dataset into 462 training, 153 validation, and 153 testing images.

The **DSB2018** source from the Data Science Bowl 2018 competition, contains 670 fluorescence-stained images with resolutions ranging from $256 \times 256$ to $520 \times 696$ pixels using DAPI and Hoechst stains. We split the dataset into 380 training, 67 validation, and 50 testing images.

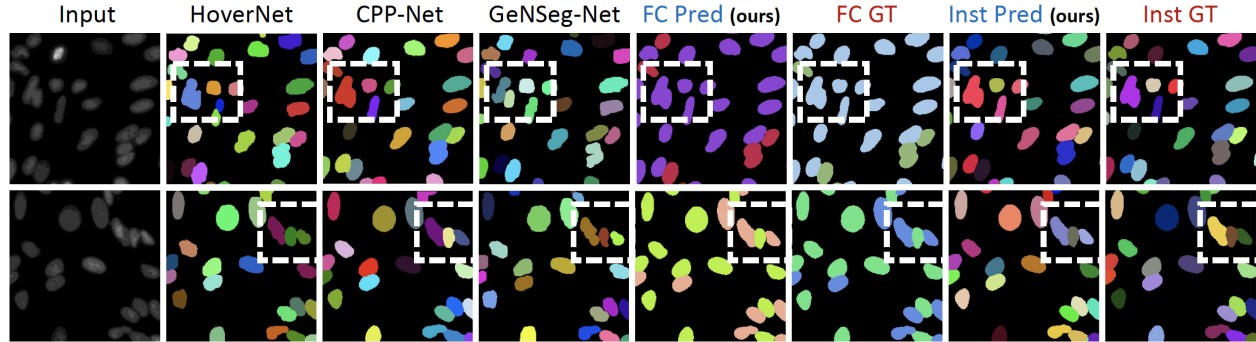

*Figure 6.* The visualization comparisons between different methods.

| Methods | Metrics | | | | |
|---|---|---|---|---|---|
| | DICE (↑) | AJI (↑) | DQ (↑) | SQ (↑) | PQ (↑) |
| DCAN (Chen et al., 2016) | 0.795 | 0.676 | 0.743 | 0.780 | 0.626 |
| HoverNet (Graham et al., 2019) | 0.898 | 0.762 | 0.863 | 0.877 | 0.762 |
| NucleiSegNet (Lal et al., 2021) | 0.904 | 0.671 | 0.784 | 0.843 | 0.682 |
| DoNet (Jiang et al., 2023) | 0.823 | 0.716 | 0.787 | 0.829 | 0.673 |
| CPP-Net (Chen et al., 2023) | 0.914 | 0.813 | 0.866 | **0.879** | 0.758 |
| GeNSeg (Xu et al., 2024) | 0.856 | 0.781 | 0.843 | 0.791 | 0.759 |
| Un-SAM (Chen et al., 2025) | 0.902 | 0.786 | 0.826 | 0.834 | 0.747 |
| CellPose (Stringer, 2025) | 0.923 | 0.824 | 0.862 | 0.871 | 0.764 |
| FCIS (Ours) | **0.939** | **0.828** | **0.875** | 0.878 | **0.770** |

*Table 2.* The comparison performances on DSB2018 dataset.

| Methods | Metrics | | | | |
|---|---|---|---|---|---|
| | DICE (↑) | AJI (↑) | DQ (↑) | SQ (↑) | PQ (↑) |
| DCAN (Chen et al., 2016) | 0.921 | 0.816 | 0.875 | 0.850 | 0.773 |
| HoverNet (Graham et al., 2019) | 0.941 | 0.891 | 0.924 | 0.911 | 0.856 |
| NucleiSegNet (Lal et al., 2021) | 0.939 | 0.671 | 0.809 | 0.844 | 0.719 |
| DoNet (Jiang et al., 2023) | 0.933 | 0.836 | 0.882 | 0.871 | 0.794 |
| CPP-Net (Chen et al., 2023) | 0.944 | 0.914 | 0.917 | 0.914 | 0.898 |
| GeNSeg-Net (Xu et al., 2024) | 0.934 | 0.907 | 0.913 | 0.911 | 0.915 |
| Un-SAM (Chen et al., 2025) | 0.933 | 0.912 | 0.909 | 0.911 | 0.904 |
| CellPose (Stringer, 2025) | 0.949 | 0.917 | 0.912 | 0.922 | 0.914 |
| FCIS (Ours) | **0.954** | **0.921** | **0.926** | **0.945** | **0.935** |

*Table 4.* The comparison performances on BBBC006v1 dataset.

| Methods | Metrics | | | | |
|---|---|---|---|---|---|
| | DICE (↑) | AJI (↑) | DQ (↑) | SQ (↑) | PQ (↑) |
| DCAN (Chen et al., 2016) | 0.778 | 0.587 | 0.659 | 0.721 | 0.506 |
| HoverNet (Graham et al., 2019) | 0.798 | 0.646 | 0.718 | 0.782 | 0.595 |
| NucleiSegNet (Lal et al., 2021) | 0.752 | 0.544 | 0.618 | 0.689 | 0.457 |
| DoNet (Jiang et al., 2023) | 0.781 | 0.612 | 0.684 | 0.750 | 0.544 |
| CPP-Net (Chen et al., 2023) | 0.814 | 0.638 | 0.711 | 0.776 | 0.583 |
| Un-SAM (Chen et al., 2025) | 0.801 | 0.629 | 0.704 | 0.767 | 0.570 |
| CellPose (Stringer, 2025) | 0.787 | 0.626 | 0.703 | 0.764 | 0.591 |
| FCIS (Ours) | **0.816** | **0.653** | **0.721** | **0.796** | **0.610** |

*Table 3.* The comparison performances on PanNuke dataset.

| Methods | Metrics | | | | |
|---|---|---|---|---|---|
| | DICE (↑) | AJI (↑) | DQ (↑) | SQ (↑) | PQ (↑) |
| DCAN (Chen et al., 2016) | 0.881 | 0.772 | 0.571 | 0.736 | 0.446 |
| HoverNet (Graham et al., 2019) | 0.907 | 0.814 | 0.602 | 0.739 | 0.445 |
| NucleiSegNet (Lal et al., 2021) | 0.874 | 0.788 | 0.583 | 0.734 | 0.439 |
| DoNet (Jiang et al., 2023) | 0.878 | 0.754 | 0.577 | 0.720 | 0.431 |
| GeNSeg-Net (Xu et al., 2024) | 0.869 | 0.747 | 0.572 | 0.722 | 0.433 |
| Un-SAM (Chen et al., 2025) | 0.904 | 0.808 | 0.597 | 0.734 | 0.442 |
| CellPose (Stringer, 2025) | 0.911 | **0.823** | **0.609** | 0.740 | 0.451 |
| FCIS (Ours) | **0.922** | 0.819 | 0.599 | **0.741** | **0.456** |

*Table 5.* The comparison performances on YeaZ dataset.

The **PanNuke** dataset includes 7901 H&E-stained images, each 256×256 pixels, originating from 19 organs, with a total of 189,744 annotated nuclei. We divide this dataset into 2656 training, 2523 validation, and 2722 testing images.

The **YeaZ** comprises 306 bright-field (BF) images with resolutions ranging from 301×301 to 1463×1311 pixels, and 43 phase-contrast (PC) images with resolutions ranging from 256×256 to 1988×2000 pixels. Due to the limited number of PC images, we merge the BF and PC datasets to train a unified model, resulting in 300 training, 20 validation, and 29 testing images.

### 5.2. Implementation Details and Evaluation Metrics

Our all experiments are conducted using PyTorch on an NVIDIA A100 GPU. We employ stochastic gradient descent (SGD) as the optimizer, with a learning rate of 0.01, momentum of 0.9, and weight decay of 0.0005. The network is trained for 200 epochs. Segmentation performance is evaluated using the DICE coefficient, Aggregated Jaccard Index (AJI) (Kumar et al., 2017), Detection Quality (DQ) (Kirillov et al., 2019), Segmentation Quality (SQ), and Panoptic Quality (PQ) metrics. In all tables presented in this paper, the **highest** performance scores are highlighted in bold, while the second-best scores are underlined.

### 5.3. Main Experiments

We evaluate the performance of our proposed method against eight state-of-the-art models across three benchmark datasets. The compared methods include the detection-based DoNet (Jiang et al., 2023); contour prediction-based approaches such as DCAN (Chen et al., 2016), NucleiSegNet (Lal et al., 2021), and GeSegNet (Xu et al., 2024); distance mapping-based methods including HoverNet (Gra-

| Settings | | | DSB2018 | | | | | Settings | | | PanNuke | | | | |
|---|---|---|---|---|---|---|---|---|---|---|---|---|---|---|---|
| | | | DICE | AJI | DQ | SQ | PQ | | | | DICE | AJI | DQ | SQ | PQ |
| Baseline | | | 0.876 | 0.751 | 0.847 | 0.856 | 0.746 | Baseline | | | 0.786 | 0.627 | 0.705 | 0.778 | 0.580 |
| w. Four-color | | | 0.843(-3.3) | 0.725(-2.6) | 0.805(-4.2) | 0.827(-2.9) | 0.674(-7.2) | w. Four-color | | | 0.766(-2.0) | 0.617(-1.0) | 0.686(-1.9) | 0.752(-2.6) | 0.559(-2.1) |
| Asymp. | Trans. | Samp. | DICE | AJI | DQ | SQ | PQ | Asymp. | Trans. | Samp. | DICE | AJI | DQ | SQ | PQ |
| ✓ | | | 0.862 | 0.740 | 0.812 | 0.831 | 0.679 | ✓ | | | 0.773 | 0.624 | 0.691 | 0.763 | 0.565 |
| ✓ | ✓ | | 0.883 | 0.756 | 0.829 | 0.844 | 0.701 | ✓ | ✓ | | 0.787 | 0.630 | 0.710 | 0.774 | 0.572 |
| | | ✓ | 0.910 | 0.785 | 0.846 | 0.863 | 0.741 | | | ✓ | 0.803 | 0.642 | 0.714 | 0.776 | 0.598 |
| ✓ | ✓ | ✓ | 0.939 | 0.828 | 0.875 | 0.878 | 0.770 | ✓ | ✓ | ✓ | 0.816 | 0.653 | 0.721 | 0.796 | 0.610 |

*Table 6.* Ablation studies on the DSB2018 and PanNuke datasets. **Baseline** denotes the binary semantic segmentation model based on U-Net (Ronneberger et al., 2015). **w. Four-color** increases the number of channels from two to five by directly employing four-color encoding in Algorithm 1 as ground truth. **Asymp.** represents the asymptotic training method, **Trans.** applies the encoding transformation method, and **Samp.** introduces a negative sampling constraint for adjacent cells.

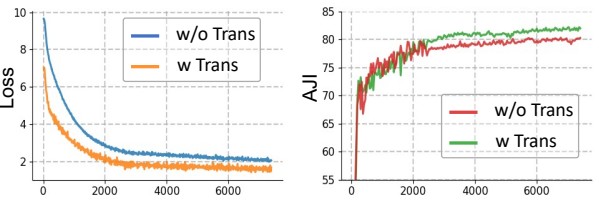

*Figure 7.* Convergence analysis of the training loss and AJI on validation set before and after applying the encoding transformation.

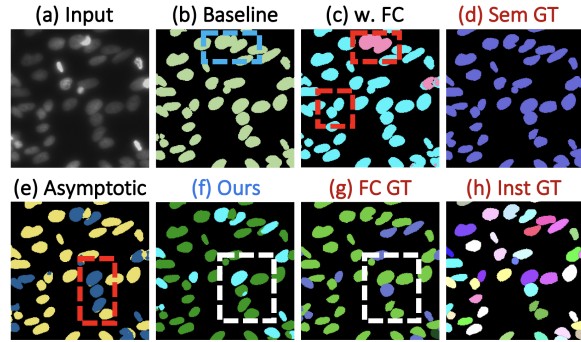

*Figure 8.* The visualization comparisons between different settings. The blue box indicates that the binary semantic prediction cannot distinguish adjacent cells. The red boxes indicate the four-color encoding lacks effective supervision for adjacent cells. The white boxes indicate our FCIS encodes adjacent cells with distinct colors.

ham et al., 2019), CPP-Net (Chen et al., 2023), and CellPose (Stringer, 2025); as well as SAM-based foundation model Un-SAM (Chen et al., 2025). Quantitative results are summarized in Tables 2–5. It is worth noting that GeSegNet, which was not designed for pathological image segmentation and performs poorly on the PanNuke dataset, is excluded from comparisons on that dataset.

From the results, we can see that FCIS consistently outperforms existing methods across all datasets. It achieves the highest DICE and AJI scores, demonstrating superior segmentation accuracy and instance-level consistency. In the DSB2018 dataset, our model achieves a DICE score of 0.939, surpassing Un-SAM and CellPose, among the best-performing prior methods. The PQ metric of 0.770 further indicates our model's ability to maintain segmentation quality and object-level distinction. In BBBC006v1, we can observe similar trends. While segmenting in the more challenging PanNuke, FCIS achieves 0.610 on the PQ, outperforming all previous methods and confirming its generalization capabilities. Although HoverNet achieves comparable performance to our method but incurs significantly higher parameter counts and computational complexity, as shown in Table 1. Therefore, considering the trade-off between model performance and computational cost, our method demonstrates a more pronounced overall advantage.

Furthermore, we visually compare different models in Figure 6. First, the results from "FC Pred" demonstrate that our method strictly adheres to the four-color encoding rule,

ensuring that adjacent cells are assigned distinct colors, enhancing instance differentiation. By comparing the cell morphologies produced by different methods, we also observe that our segmentation results align more closely with the ground truth. This improvement can be attributed to incorporating the negative sampling learning method, effectively enhancing the boundary delineation. Additionally, due to page constraints, more visualization results are provided in the Supplementary Material.

### 5.4. Ablation Studies

**Effectiveness Analysis of the Method Designs**

We conduct an ablation study to evaluate the module's performance under various configurations. The ablation methods include employing an asymptotic training strategy, applying encoding transformations to the network's predictions, and introducing a sampling constraint for adjacent cells. The experimental results are presented in Table 6. From the table, we observe the following: (1) When using the four-color encoding as supervision, the model performance decreases significantly, indicating the inherent

challenges of directly employing this encoding as a training signal. (2) Adding the asymptotic training strategy or encoding transformation methods leads to slight performance improvements, suggesting that these techniques provide some regularization benefits to the learning process. (3) Introducing the sampling constraint for adjacent cells results in a substantial performance boost, highlighting the effectiveness of this design in enforcing spatial consistency among predictions. These findings demonstrate that the proposed designs contribute positively to model performance.

### Analysis of Training Convergence

We analyze the model's convergence behavior by comparing the training loss and the validation AJI before and after applying the encoding transformation, as illustrated in Figure 7. The results indicate that incorporating the encoding transformation accelerates the convergence of the training loss, leading to faster stabilization with lower loss. Additionally, the AJI metric shows a significant improvement after applying the transformation, demonstrating the effectiveness of this design in enhancing model performance.

### Visualization of Different Settings

Based on the experimental results in Table 6, we conduct the visualization comparisons as shown in Figure 8. The red annotations represent the binary semantic segmentation ground truth (GT), the four-color encoding GT, and the instance segmentation GT, respectively. First, from the baseline results (b), it is evident that using only dual-channel predictions fails to distinguish adjacent cells effectively. Second, when directly using four-color encoding (b) as a supervision, the model lacks awareness of encoding inconsistency for adjacent cells, resulting in not only indistinguishable instances, but also fragmented predictions. By incorporating the asymptotic training (c) strategy, these issues are partially alleviated; however, distinguishing adjacent cells remains challenging. In contrast, our proposed method (f) demonstrates that the predicted results ensure not only that adjacent cells are encoded with different colors but also that the structural integrity of each instance.

## 6. Conclusions

We present a novel approach to cell instance segmentation by leveraging the four-color theorem, which reformulates the instance segmentation problem as a four-class semantic segmentation task. This transformation significantly reduces computational overhead and simplifies model design. To address challenges arising from the non-uniqueness of color encodings, we propose an asymptotic training strategy and an encoding transformation mechanism that ensure stable optimization. Extensive experiments on diverse biomedical imaging modalities, including fluorescence, H&E, and bright-field microscopy, demonstrate that our method con-

sistently achieves superior segmentation accuracy and efficiency compared to state-of-the-art approaches. Future work will explore adaptive encoding strategies that dynamically respond to varying tissue architectures and cell densities, further improving generalization across datasets. Additionally, extending the proposed framework to downstream tasks such as cell nucleus classification represents a promising research direction.

## Acknowledgments

This research was supported by the National Key R&D Program of China (No. 2023YFC3305600), the Federal Ministry of Education and Research in Germany under funding reference 161L0272, and the Ministry of Culture and Science of the State of North Rhine-Westphalia. The authors would like to thank the anonymous reviewers for their valuable comments and suggestions, which greatly improved the quality of this paper.

## Impact Statement

This paper introduces a novel cell instance segmentation method based on four color theorem that significantly improves the accuracy and computational efficiency. By simplifying the segmentation task, this approach has the potential to enhance automated biomedical image analysis and accelerate clinical diagnostics.

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

## Supplementary Material

## A. Data Splitting

We applied an overlapping cropping method to the DSB2018, BBBC006V1, and YeaZ datasets during the data preprocessing stage. Specifically, we used a sliding window with a stride of 128 to extract 256×256 patches. Since the original image size in the PanNuke dataset is already 256×256, no additional processing was required. The number of samples in the training, validation, and test sets is shown in Table 7.

| Datasets | No. Traing | No. Valiadation | No. Testing |
|----------|-----------|-----------------|-------------|
| DSB2018 | 602 | 109 | 89 |
| PanNuke | 2656 | 2522 | 2722 |
| BBBC006v1 | 1848 | 612 | 612 |
| YeaZ | 1000 | 140 | 200 |

*Table 7.* The number of samples in the training, validation, and test datasets.

## B. Related Work

### B.1. Detection-Based Cell Segmentation

The challenge of distinguishing individual cells in overlapping regions has long been a critical issue in instance segmentation. With the introduction of Faster R-CNN Ren et al. (2016), Mask R-CNN He et al. (2017) extended this detection framework by incorporating an instance segmentation module. This two-stage approach first generates bounding boxes to locate individual instances and then performs segmentation within these regions. Mask R-CNN's inherent ability to separate instances without requiring complex post-processing has made it a widely adopted framework for semi-supervised cell instance segmentation tasks Zhou et al. (2020).

### B.2. Contour Prediction-Based Cell Segmentation

Contour-based segmentation methods focus on explicitly predicting cell boundaries to achieve instance separation. Early works, such as U-Net Ronneberger et al. (2015), facilitated boundary learning by assigning higher pixel-wise weights to cell edges, followed by post-processing techniques like watershed or contour detection to delineate individual instances. This architecture significantly influenced deep learning-based segmentation, particularly in medical imaging. Subsequent advancements introduced explicit contour prediction to improve instance separation. DCAN Chen et al. (2016) incorporated additional semantic categories for boundary pixels, enabling clearer differentiation between cells and the background. UNet++ Zhou et al. (2018) refined U-Net's performance by employing nested skip connections, while FullNet Qu et al. (2019) and CIA-Net Zhou et al. (2019) leveraged multi-scale

context aggregation to enhance boundary delineation. More recent models, such as TSFD-Net Ilyas et al. (2022) and GeNSeg-Net Xu et al. (2024), continue to advance the field by integrating sophisticated architectures designed to improve boundary prediction accuracy.

### B.3. Distance-Based Cell Segmentation

Distance-based segmentation approaches predict spatial relationships between pixels and their corresponding cell instances, facilitating robust separation of adjacent cells. StarDist Schmidt et al. (2018), one of the pioneering methods in this category, introduced radial distance predictions, which proved effective for segmenting cells with irregular shapes. HoverNet Graham et al. (2019) extended this concept by simultaneously predicting a distance map and a classification map, enabling accurate instance separation in densely packed regions. CDNet He et al. (2021) further improved generalization across datasets by employing multi-task learning. Recent advancements have explored more sophisticated architectures to enhance both segmentation accuracy and computational efficiency. SONNET Doan et al. (2022) introduced a self-organizing network to model complex spatial relationships, while TransUNet He et al. (2023) combined transformer-based architectures with distance prediction to enhance feature representation. CPP-Net Chen et al. (2023) and SMILE Pan et al. (2023) incorporated context-aware modules to improve adaptability to diverse cell morphologies. Emerging models such as CellViT Hörst et al. (2024) and RepSNet Xiong et al. (2025) integrate vision transformers with structural priors, further advancing distance-based segmentation techniques for challenging datasets.

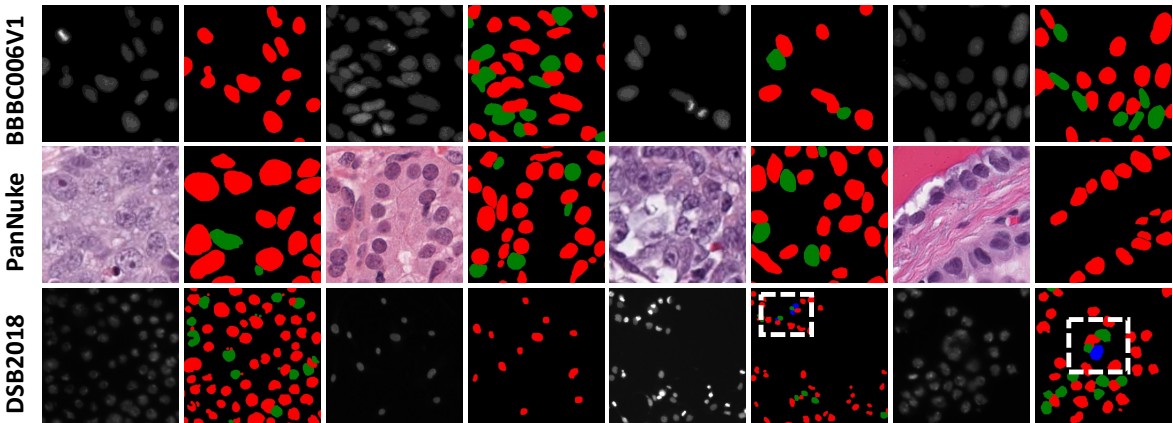

*Figure 9.* Visualization of four-color encoding results.

## C. The Analysis of Four-color Encoding

The four-color encoding method highlights the feasibility of transforming cell instance segmentation into a semantic segmentation task. To better understand the characteristics of four-color encoding, we present visualizations of encoded images from multiple datasets, as shown in Figure 9. In this figure, we randomly selected images from three different datasets and applied four-color encoding. The results reveal the following patterns:

(1) The majority of cells are encoded in red, while a smaller proportion are assigned green;

(2) Cells encoded in blue are scarce, appearing only in highly dense regions (highlighted by white box), typically with one or two occurrences;

(3) The fourth encoding category (represented by yellow) does not appear, indicating that cell encoding is more constrained and simplified than the traditional map-coloring problem.

Furthermore, the statistical analysis of the four-color encoding results, illustrated in Figure 10, aligns with the observed distribution of cell color assignments, further validating the characteristics of this encoding approach.

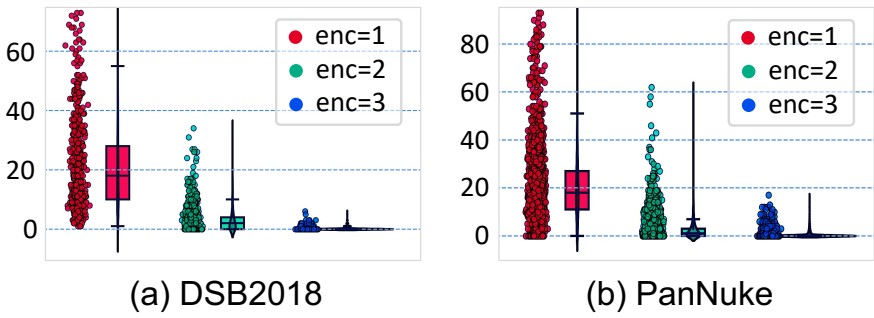

(a) DSB2018        (b) PanNuke

*Figure 10.* Statistics of different color encodings

## D. Preliminaries

We provide essential definitions and concepts to establish the foundation for the proposed method.

**Definition 1. Undirected Graph.** An undirected graph is represented as $G = (V, E)$, where $V$ is the set of vertices, and $E$ is the set of edges. An edge $e = (u, v) \in E$ indicates that vertices $u$ and $v$ are adjacent.

**Definition 2. Coloring Number.** The chromatic number of a graph $G$, denoted as $\chi(G)$, is the minimum number of colors required to color the vertices of $G$ such that no two adjacent vertices share the same color.

**Definition 3. Maximum Degree.** The degree of a vertex $v \in V$, denoted as $d(v)$, is the number of vertices adjacent to $v$. The maximum degree of the graph $G$ is defined as $\Delta(G) = \max_{v \in V} d(v)$.

**Definition 4. Chain Structure:** A type of graph where the vertices are arranged in a linear path, formally known as a path graph $P_n$. In the structure, each vertex is connected to at most two adjacent vertices. For instance, in the graph $P_4$ with 4 vertices, the coloring sequence can be described as:

$$v_1 \rightarrow \text{color 1}, \ v_2 \rightarrow \text{color 2}, \ v_3 \rightarrow \text{color 1}, \ v_4 \rightarrow \text{color 2}.$$

**Definition 5. Rectangular Structure:** A rectangular structure is a graph where vertices are arranged in a regular rectangular grid. Such graphs are a specific type of planar graph, where each vertex typically

has a degree of 2 or 4, satisfying $\Delta(G) \leq 4$.

**Definition 6. Planar Graph.** A planar graph is a graph that can be embedded in the plane such that no edges intersect. According to the Four-Color Theorem, the chromatic number of a planar graph satisfies $\chi(G) \leq 4$.

## E. Theorem and Proof

**Theorem 1. Global Optimality of Greedy Coloring**: Let $G = (V, E)$ be an undirected graph representing a cell distribution, where $V$ is the set of vertices (cells), and $E$ is the set of edges representing adjacency relationships between cells. Suppose $G$ satisfies the following conditions:
(1) $G$ is a planar graph, meaning it can be embedded in a plane such that no two edges intersect;
(2) The maximum degree of $G$, denoted by $\Delta(G)$, satisfies:

$$\Delta(G) \leq k, \quad \text{where } k \leq 4;$$

(3) The vertex distribution of $G$ follows either a chain structure (a path graph $P_n$) or a rectangular structure (a grid-like planar graph).

Then, the chromatic number of $G$, defined as the minimum number of colors required to color the vertices such that no two adjacent vertices share the same color, satisfies:

$$\chi_{\text{greedy}}(G) = \chi(G).$$

Where $\chi_{\text{greedy}}(G)$ is the coloring number by applying the greedy algorithm with any arbitrary vertex ordering. This result demonstrates that the greedy algorithm produces a globally optimal solution to the graph coloring problem.

**Proof 1:**

**Definition and Properties of Greedy Algorithm:** The greedy algorithm colors graph $G$ as follows: - Traverse all vertices in the order $v_1, v_2, \ldots, v_n$; - For each vertex $v_i \in V$, assign the smallest color that has not been used by any of its adjacent vertices; - Each vertex checks at most $\Delta(G)$ adjacent vertices, and the number of colors needed is at most $\Delta(G) + 1$.

Thus, the chromatic number generated by the greedy algorithm satisfies:

$$\chi_{\text{greedy}}(G) \leq \Delta(G) + 1$$

**Optimality Analysis under Special Structures:**

(a) Chain Structure (Path Graph $P_n$): For a path graph $P_n$, each vertex has a degree $\Delta(P_n) = 2$. - The chromatic number of a path graph is $\chi(P_n) = 2$; - When the greedy algorithm colors in any vertex

order, it uses at most two colors:

$$\chi_{\text{greedy}}(P_n) = \chi(P_n) = 2$$

Therefore, the greedy algorithm is optimal for path graphs.

(b) Rectangular Structure: For cells arranged in a rectangular grid, graph $G$ is planar, and $\Delta(G) \leq 4$. According to the Four Color Theorem:

$$\chi(G) \leq 4$$

The greedy algorithm, in each iteration, uses the smallest available color, and each vertex checks at most 4 adjacent vertices. Therefore, the chromatic number generated by the greedy algorithm satisfies:

$$\chi_{\text{greedy}}(G) \leq 4 = \chi(G)$$

Thus, the greedy algorithm is also optimal for rectangular structures. Extending Local Optimality to Global Optimality:

**Local Sparsity:** Due to the distribution properties of graph $G$, in locally clustered regions, the number of vertices is limited and the maximum degree is low. Hence, the greedy algorithm is optimal in local regions.

**Global Sparsity of Planar Graphs:** The global distribution of planar graphs is sparse, and edges connecting different regions are limited, causing little interference with the local optimal solution. As a result, the local optimality of the greedy algorithm extends to global optimality.

Based on the above analysis, the chromatic number of graph $G$, which satisfies the given conditions, is equal to the minimum chromatic number:

$$\chi_{\text{greedy}}(G) = \chi(G)$$

Thus, the greedy algorithm is an effective method for generating the minimum color coding in this scenario.

**Theorem 2. Greedy Coloring Compatibility:** In the cell instance segmentation task, let the encoding matrix generated by the greedy algorithm be:

$$\mathbf{C} \in \mathbb{R}^{n \times k}, (k \leq 4). \tag{16}$$

And the encoding matrix predicted by the network is:

$$\mathbf{P} \in \mathbb{R}^{n \times k'}, \tag{17}$$

where $n$ represents the number of cells, $k$ is the number of colors used in the greedy algorithm, and $k'$ is the number of predicted encodings. If the predicted encoding matrix $\mathbf{P}$ has one of the relations with

the greedy encoding, i.e., substitution, exchange, modification of rules. Then there exists a mapping function:

$$f : \mathbf{P} \to \mathbf{C}, \tag{18}$$

such that the network's predicted result $\mathbf{P}$ can be transformed into the four-color encoding result $\mathbf{C}$.

**Proof 2:**

The four-color encoding matrix $\mathbf{C}$ generated by the greedy algorithm satisfies the following properties:

(a) Sparsity: Each row has at most one nonzero element ($\mathbf{C}[i,j] \in \{0,1\}$), representing that the $i$-th node uses the $j$-th color;

(b) Optimality: The number of colors used is minimized, $\mathrm{rank}(\mathbf{C}) = k$, and $k \leq 4$;

(c) Adjacency constraint: Any two adjacent nodes $(v_i, v_j)$ satisfy $\mathbf{C}[i,:] \neq \mathbf{C}[j,:]$ (i.e., they cannot use the same color).

These properties can be formally expressed as follows:

(1) Sparsity: $\sum_{j=1}^{k} \mathbf{C}[i,j] = 1, \forall i$. (2) Adjacency constraint: If $e_{i,j} = 1$, then $\mathbf{C}[i,:] \cdot \mathbf{C}[j,:]^{\top} = 0$.

The encoding matrix $\mathbf{P} \in \mathbb{R}^{n \times k'}$ predicted by the network exhibit non-uniqueness due to the following reasons:

(a) Substitution: Some rows of the encoding are replaced, introducing redundancy;

(b) Exchange: The order of the columns is changed;

(c) Rule modification: Additional colors are introduced, resulting in $k' > k$.

Thus, the column rank of $\mathbf{P}$ satisfies:

$$\mathrm{rank}(\mathbf{P}) \geq k. \tag{19}$$

Hence, we need to construct a mapping function $f : \mathbf{P} \to \mathbf{C}$ to transform the predicted encoding matrix $\mathbf{P}$ into the four-color encoding matrix $\mathbf{C}$ that satisfies the constraints.

(1) Column Redundancy Elimination

A linear transformation is applied to eliminate redundant columns in $\mathbf{P}$, ensuring that the resulting matrix has rank $k$. Specifically: Define a column transformation matrix $\mathbf{T} \in \mathbb{R}^{k' \times k}$, where

$$\mathbf{T} = \underset{\mathbf{T}}{\arg\min} \|\mathbf{PT} - \mathbf{C}\|_F^2, \quad \text{s.t. } \mathrm{rank}(\mathbf{PT}) = k.$$

The transformed matrix is

$$\mathbf{P}' = \mathbf{PT},$$

where $\mathbf{P}' \in \mathbb{R}^{n \times k}$, and $\mathrm{rank}(\mathbf{P}') = k$.

(2) Column Order Adjustment

The columns of $\mathbf{P}'$ are reordered to align with the column order of $\mathbf{C}$. Let the column permutation matrix be $\mathbf{S} \in \mathbb{R}^{k \times k}$, then

$$\mathbf{C} = \mathbf{P}'\mathbf{S}.$$

The matrix $\mathbf{S}$ is a permutation matrix satisfying $\mathbf{S}^\top \mathbf{S} = \mathbf{I}$.

(3) Adjacency Constraint Verification

After the mapping, the adjacency constraint is verified to ensure that the resulting matrix satisfies the four-color encoding rule:

$$\mathbf{C}[i,:] \cdot \mathbf{C}[j,:]^\top = 0, \quad \forall e_{i,j} = 1.$$

It can be seen that, for any predicted matrix $\mathbf{P}$, the three-step mapping function $f$ ensures that the transformed matrix $\mathbf{C}$ satisfies:

(1) The rank of the transformed matrix is $k$, i.e., $\mathrm{rank}(\mathbf{P}') = k$;

(2) The column order is aligned with $\mathbf{C}$;

(3) The adjacency constraint holds, making $\mathbf{C}$ a valid four-color encoding result.

Therefore, we design encoding transformation and orthogonal constraints to ensure the rationality of four-color prediction.

## F. Hyper-parameter Ablation Experiments

We conducted an ablation study on hyperparameter selection using the DSB2018 and BBBC006v1 datasets, focusing on the impact of the sampling rate and the weight of the orthogonal constraint loss function. The experimental results are presented in Tables 8 and 9.

The results indicate that increasing the sampling rate generally improves model performance. However, the performance gain from 0.5 to 0.7 is less significant than the improvement observed when increasing the sampling rate from 0.3 to 0.5. We set the sampling rate to 0.5 in the main experiments to balance model performance and computational efficiency.

Furthermore, we examined the effect of the orthogonal constraint loss weight on model performance. A significant performance drop is observed when the weight is set to 1. We hypothesize that this is due to the insufficient enforcement of the orthogonal constraint at lower weights, reducing the model's ability to distinguish adjacent instances and ultimately degrading segmentation performance effectively.

| Ratio | DSB2018 | | | | | Ratio | BBBC006v1 | | | | |
|---|---|---|---|---|---|---|---|---|---|---|---|
| | DICE | AJI | DQ | SQ | PQ | | DICE | AJI | DQ | SQ | PQ |
| $r = 0.3$ | 0.913 | 0.803 | 0.854 | 0.871 | 0.758 | $r = 0.3$ | 0.947 | 0.917 | 0.893 | 0.926 | 0.899 |
| $r = 0.5$ | 0.939 | 0.828 | 0.875 | 0.878 | 0.770 | $r = 0.5$ | 0.954 | 0.921 | 0.926 | 0.945 | 0.935 |
| $r = 0.7$ | 0.941 | 0.832 | 0.866 | 0.881 | 0.779 | $r = 0.7$ | 0.946 | 0.924 | 0.933 | 0.951 | 0.938 |

*Table 8.* Ablation studies of sampling ration on DSB2018 and BBBC006v1 datasets.

| Weight | DSB2018 | | | | | Weight | BBBC006v1 | | | | |
|---|---|---|---|---|---|---|---|---|---|---|---|
| | DICE | AJI | DQ | SQ | PQ | | DICE | AJI | DQ | SQ | PQ |
| $\lambda = 1$ | 0.908 | 0.798 | 0.832 | 0.825 | 0.716 | $\lambda = 1$ | 0.922 | 0.898 | 0.891 | 0.904 | 0.880 |
| $\lambda = 2$ | 0.939 | 0.828 | 0.875 | 0.878 | 0.770 | $\lambda = 2$ | 0.954 | 0.921 | 0.926 | 0.945 | 0.935 |

*Table 9.* Ablation studies of weight setting on DSB2018 and BBBC006v1 datasets.

## G. More Visualization Results

We present the semantic and instance segmentation results, including error analysis, as shown below. Specifically, the subfigures include the input image, pixel-wise error analysis, four-class semantic ground truth, and instance segmentation labels (the last two subfigures can be ignored). The results in the DSB2018 and BBBC006v1 datasets demonstrate that our method not only achieves accurate instance segmentation but also excels in pixel-wise classification by significantly reducing false positive (FP) and false negative (FN) prediction errors. These results validate the effectiveness of our approach.

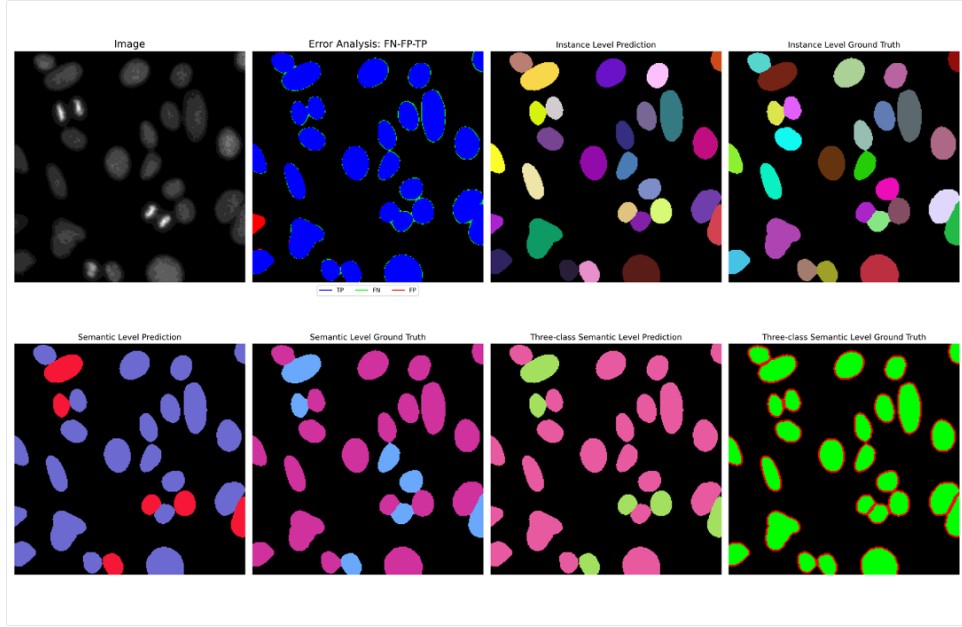

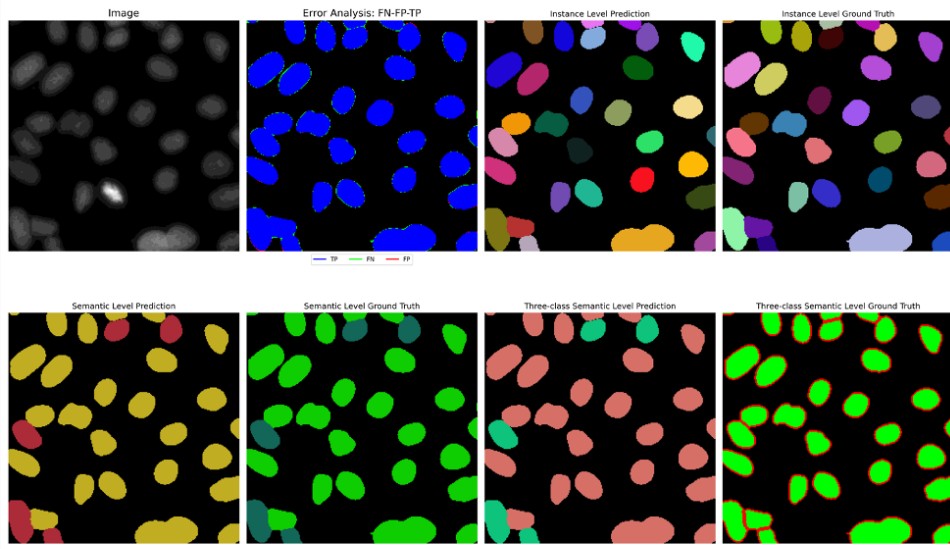

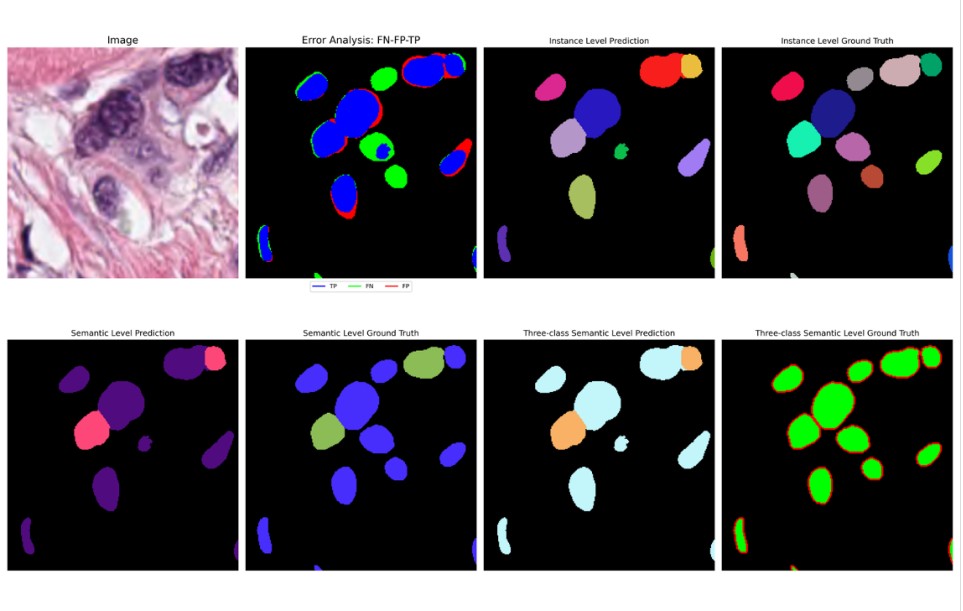

## H. Others

To better demonstrate the rationality of our model's module design, we plot the convergence curves of various loss functions during the training process on the PanNuke dataset, as shown in Figure 11. The results indicate that our method ensures stable model convergence.

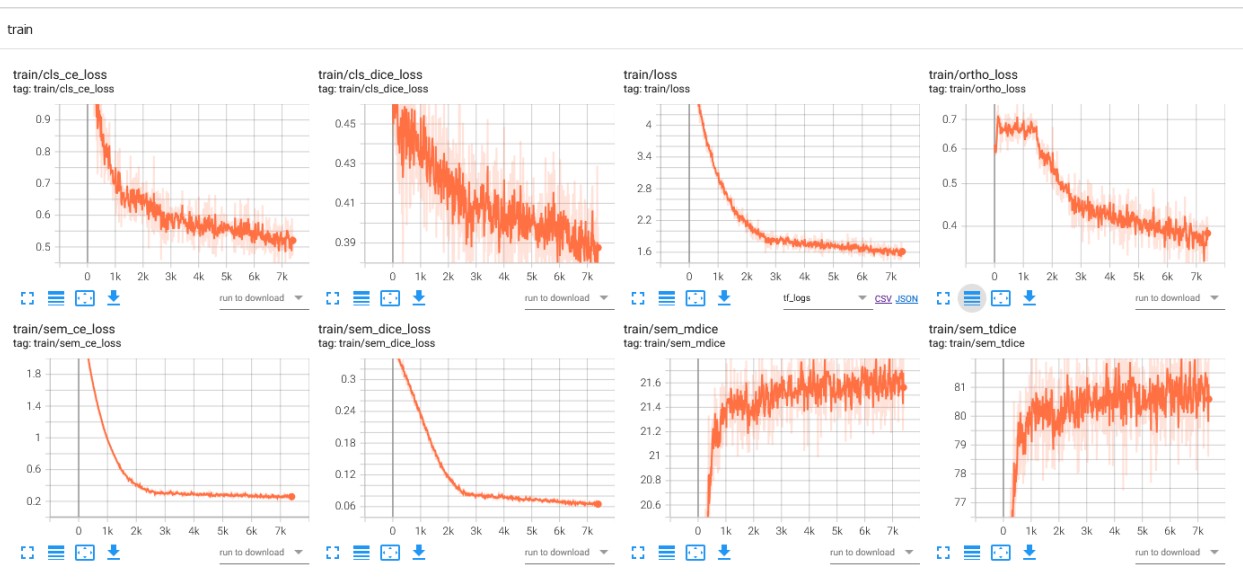

*Figure 11.* The convergence of loss function and Dice in training process.

