# OpenReview forum: "The Four Color Theorem for Cell Instance Segmentation"
_ICML.cc/2025/Conference — ICML 2025 poster_

### Official Review · Reviewer_YyUg · 2025-03-11

**Overall Recommendation:** 3

**Summary:**

The paper presents a novel cell instance segmentation method inspired by the four-color theorem, which simplifies the segmentation task by transforming it into a four-class semantic segmentation problem. The key contributions include:

Four-Color Encoding Scheme: Cells are treated as "countries" and tissues as "oceans," allowing the use of a four-color encoding to ensure adjacent cells receive distinct labels.

Asymptotic Training Strategy: This step-by-step approach first distinguishes foreground and background, then predicts categories within the foreground, addressing class imbalance.
Encoding Transformation Method: This method maps the predicted output to a consistent four-color representation, ensuring training stability.

Theoretical Analysis: The authors prove the global optimality of the greedy algorithm for cell coloring under specific conditions.
Empirical Validation: The method achieves state-of-the-art performance on three datasets (BBBC006v1, DSB2018, PanNuke), demonstrating robustness and reduced computational complexity.

The proposed method effectively balances performance and efficiency, offering a promising solution for biomedical image segmentation.

## update after rebuttal
The responses are satisfactory, and we acknowledge the improvements made in both theoretical and empirical aspects.

**Claims And Evidence:**

The claims made in the submission are largely supported by clear and convincing evidence. The authors present a novel approach to cell instance segmentation using the four-color theorem, which is well-explained and validated through extensive experiments. The four-color encoding scheme is effectively demonstrated to transform the segmentation task into a simpler semantic segmentation problem, and the asymptotic training strategy is shown to address class imbalance issues. The encoding transformation method is theoretically justified and empirically proven to enhance model convergence and performance. The state-of-the-art performance on three datasets (BBBC006v1, DSB2018, and PanNuke) further substantiates the effectiveness of the proposed method.

However, there are certain aspects that could benefit from additional clarification. The generalizability of the method across diverse cell types and imaging modalities is not fully explored, and further discussion or experiments would be valuable to demonstrate its broader applicability. Additionally, while the impact of the non-uniqueness of encoding is addressed through the encoding transformation method, a more detailed analysis of its effect on model performance would strengthen the paper. The computational efficiency claims, though supported by a comparison of computational complexity, could be further bolstered by detailed benchmarks on standard hardware configurations. Lastly, the theoretical analysis of the global optimality of the greedy algorithm, while convincing, relies on specific assumptions about cell distribution and structure, which could be more thoroughly discussed in the context of real-world biomedical images. Overall, the submission presents a robust and innovative approach, with minor areas for enhancement to fully address potential limitations.

**Essential References Not Discussed:**

In my opinion, there are no essential related works missing from the paper that would significantly alter the understanding of its key contributions. The authors have adequately covered the relevant literature in the fields of biomedical image segmentation, graph theory, and deep learning. The application of the four-color theorem is a unique approach that addresses specific challenges in cell segmentation, and the paper provides sufficient context for its contributions.

**Experimental Designs Or Analyses:**

The experimental designs and analyses in the paper are sound and appropriate for evaluating the proposed cell instance segmentation method. The authors used three diverse datasets and standard metrics to comprehensively assess performance, demonstrating robustness and superiority over state-of-the-art methods. Ablation studies and convergence analysis further validate the effectiveness of the proposed techniques.

**Methods And Evaluation Criteria:**

The proposed methods and evaluation criteria are well-suited for cell instance segmentation in biomedical images. The four-color encoding scheme simplifies segmentation by transforming it into a four-class semantic task, while the asymptotic training strategy addresses class imbalance. The encoding transformation method ensures training stability by handling non-uniqueness in encoding. The negative sampling constraint enhances the model's ability to distinguish adjacent cells. The evaluation on three diverse datasets (BBBC006v1, DSB2018, PanNuke) using standard metrics (DICE, AJI, DQ, SQ, PQ) validates the method's robustness and generalizability. Overall, the approach effectively addresses key challenges in cell segmentation with strong empirical support.

**Other Comments Or Suggestions:**

In my opinion, the summary part of the paper is relatively small, and the innovation, experiment, advantages and disadvantages of the paper should be sorted out. However, there does not seem to be enough room for this, and I suggest that the pseudocode be placed in additional material in order to expand the conclusions.

**Other Strengths And Weaknesses:**

The paper presents an innovative application of the four-color theorem to cell instance segmentation, offering a novel perspective on a challenging problem in biomedical imaging. The transformation of the segmentation task into a four-class semantic segmentation problem is a significant contribution, simplifying the process and reducing computational complexity. The theoretical analysis and empirical validation on multiple datasets further strengthen the paper's credibility.

However, the generalizability of the method to diverse imaging modalities and complex cell structures remains unexplored. The reliance on accurate initial segmentation and the need for post-processing to correct errors also pose limitations. Addressing these issues could enhance the robustness and applicability of the proposed approach. Overall, the paper's originality and potential impact are noteworthy, but further work is needed to address its limitations.

**Questions For Authors:**

I have no other questions here.

**Relation To Broader Scientific Literature:**

The key contributions of the paper are well-aligned with the broader scientific literature, particularly in biomedical image segmentation and graph theory. The use of the four-color theorem is a novel adaptation of a classic graph theory result to address the challenge of distinguishing adjacent cells in segmentation tasks. This approach simplifies the problem by converting it into a four-class semantic segmentation task, which is a significant departure from traditional methods that often struggle with computational complexity and accuracy in high-density cell environments.

The asymptotic training strategy and encoding transformation method build on existing deep learning techniques to address training instability and non-uniqueness of encoding, which are common issues in semantic segmentation. These strategies are innovative solutions that enhance model robustness and performance, aligning with ongoing research efforts to improve segmentation accuracy and efficiency.

The theoretical analysis provided in the paper, which proves the global optimality of the greedy algorithm under specific conditions, contributes to the understanding of graph coloring in biomedical images. This analysis is grounded in established principles of graph theory and combinatorial optimization, providing a rigorous foundation for the proposed method.

Overall, the paper integrates well with the broader literature by offering a new perspective on a longstanding problem in biomedical imaging, leveraging both theoretical insights from graph theory and practical advancements in deep learning.

**Theoretical Claims:**

As a whole, there are no logical errors in the algorithm mentioned in the article, but I am concerned about the following:
This strategy relies on accurate initial segmentation. If there are errors in the initial segmentation (such as cell adhesion is not segmented correctly), the four-color strategy may not be able to effectively distinguish instances and affect the final result.
On large-scale high-resolution pathological images, the computational complexity of the four-color strategy is high, especially when constructing the adjacency graph and optimizing the coloring, which may affect the reasoning efficiency.

---

> ### Author Rebuttal · Authors · 2025-03-30
>
> We sincerely thank the reviewer for the thorough evaluation of our manuscript and the many constructive comments provided. We are also profoundly grateful for your recognition of our proposed method's innovation, theoretical contributions, and experimental design. According to your metioned several minor concerns, **we have summarized these into following five points and  made one-by-one responses below**. We hope our responses further reinforce your confidence in our approach. For your convenience, we have prepared supplementary materials to assist with your review: [Supp](https://drive.google.com/file/d/1dERoIrcnDklZhwcJDs5zXBBAFn7-I5u3/view?usp=sharing).
>
> **1. Generalizability to Diverse Imaging Modalities**
>
> Thank you for your valuable suggestions to improve our experimental comparsions, hence we conducted additional experiments on three challenging cell segmentation datasets to demonstrate broader applicability, including **fluorescence image**, **bright-field image** and **phase-contrast image**, and one **natural-scene dataset (PerSense)**. These four datasets contain many complex, low-contrast, and densely packed objects, as shown in **Supp.RG-Fig.1**.
>
> At the same time, we add extensive comparison experiments, and the quantitative results are shown in **Supp.RG-Tab.1**. From the table, we can see that our method shows comparable results across these diverse scenarios. Notably, it achieves **AJI scores of 0.834 and 0.822** on the **Yeaz-BF** and **Yeaz-PC** datasets. On the PerSense dataset, our method outperforms the second-best method by more than **2%**, demonstrating excellent generalization beyond biomedical domains and reinforcing robustness and scalability. Besides, the visualization results in **Supp.RG-Fig.2–6** further indicates our FCIS can tackle complex scenario cell segmentation and densely distributed objects segmentation.
>
>  **2. Effect of Non-Uniqueness in Encoding**
>
> As detailed in the main paper and further illustrated in **Supp.R1-Fig. 6 and Fig.7**, inconsistent color assignments across training batches can cause unstable optimization and fragmented predictions. To resolve this, we introduce an encoding transformation mechanism that includes a "buffer region" in the prediction space. This mechanism maps all variant encodings to a canonical and stable representation. As shown in **Supp.R1-Fig.7(a)**, this transformation significantly improves training convergence.
>
> To further support this claim, we provide visual comparisons in **Supp.R1-Fig.7(b)**. The segmentation results clearly demonstrate that the encoding transformation mechanism eliminates fragmented predictions, which confirms its effectiveness in stabilizing training and enhancing model robustness.
>
>  **3. Computational Efficiency and Hardware Benchmarks**
>
> Thank you for your valuable suggestion regarding hardware dependency in our computational complexity analysis. In the initial submission, we used an **NVIDIA A100 GPU** to compute  parameter quantification and FLOPs. We will clearly state this hardware configuration in the revised manuscript.
>
> **4. Theoretical Assumptions and Real-World Applicability**
>
> As you rightly pointed out, our method is based on certain assumptions about cell distribution characteristics—such as local clustering and global dispersion, which imply planar distribution and non-cross instances. These assumptions hold in our current setting and ensure the feasibility of constructing a valid four-color adjacency graph. However, as you suggested, extending our method to more complex scenarios that do not meet these assumptions would require further theoretical analysis and algorithmic adaptation. We will continue to explore and refine this direction in our future work.
>
> Thank you again for your insightful and constructive suggestion.
>
> **5. Summary and Pseudocode Placement**
>
> Thank you for noting the brevity of the summary section. However, due to it is currently not allow to update the main text, we promise to expand the conclusion to better highlight the key contributions, main results, and limitations in a future version.
>
> Meantime, we also appreciate your suggestion to move the pseudocode to the supplementary material to improve the structure and readability of the main paper, and we will revise it accordingly. We appreciate this suggestion, which will help improve the overall presentation of the paper.
>
> Once again, we sincerely thank you for your thoughtful feedback and positive evaluation. We hope the additional experiments, theoretical clarifications, and manuscript revisions have addressed your concerns.

---

> > ### Comment · Reviewer_YyUg · 2025-04-04
> >
> > Thank you for your detailed and thoughtful rebuttal. The responses are satisfactory, and we acknowledge the improvements made in both theoretical and empirical aspects.

---

> > > ### Author Response · Authors · 2025-04-04
> > >
> > > Thank you for your continued guidance and for recognizing our responses during the rebuttal phase. We are truly encouraged by your acknowledgment of the improvements we have made. We will continue to do our best to advance the field of instance segmentation. Once again, thank you for your support and encouragement.

---

### Official Review · Reviewer_RU3q · 2025-03-13

**Overall Recommendation:** 3

**Summary:**

This paper introduces a new approach to cell instance segmentation based on the Four-Color Theorem from graph theory. The authors propose reformulating instance segmentation as a constrained semantic segmentation problem using only four classes. This method simplifies the task by ensuring that adjacent cells receive different labels, allowing instance differentiation without explicit instance segmentation. Evaluated on three biomedical image datasets

## Update after rebuttal
I have adjusted my score to reflect the improvements. Please see my reply to the rebuttal below.

**Claims And Evidence:**

- Instead of detecting individual cell instances explicitly, the method treats segmentation as a semantic task where each foreground cell is assigned one of four colors to ensure instance differentiation. The authors present a greedy four-color encoding algorithm (Algorithm 1) that assigns colors while ensuring that adjacent cells have distinct labels. They demonstrate that this reduces model complexity compared to detection- and contour-based approaches.

- Directly training on four-color encoding leads to instability due to class imbalance and non-unique encoding. Authors propose binary foreground-background segmentation and four-class label assignment within the foreground. An ablation study shows that using four-color encoding directly degrades performance. The asymptotic training method significantly improves performance metrics.

**Essential References Not Discussed:**

The authors compare their model mostly against contour-based, detection-based, and distance-based methods (HoverNet, DoNet, CPP-Net, etc.). Although Cellpose may not fall into these categories, it is widely regarded as one of the most robust and generalist cell segmentation models, capable of handling touching and overlapping cells across diverse datasets. The fact that this paper only briefly mentions it and does not include it in the benchmarks raises serious concerns about completeness of experimental validation. It is very fast and works for any type of images (H&E, cryo-EM, TEM, confocal, bright field etc.). It is only cited once in this sentence "The advancement of deep learning revolutionized automated cell segmentation ...". There are three versions of CellPose already, most recent of which is recently published. The version cited is from 2021.

**Experimental Designs Or Analyses:**

- The experimental design is structured to compare the four-color method against existing cell segmentation approaches. However, some choices may bias the results or leave important questions unanswered. The datasets do not include highly overlapping cell populations. So, the methods generalizability is in question.

- Cellpose, a leading generalist model for cell segmentation, is missing in the comparison. This is major gap since Cellpose is robust across various imaging conditions.

- Ablations studies support the paper's claims and is a strong point of the experimental design.

**Methods And Evaluation Criteria:**

The proposed method and evaluation criteria generally make sense for the problem of cell instance segmentation in biomedical imaging. However, there are some important considerations regarding their real-world applicability. The method assumes that cells are adjacent but not significantly overlapping. In cases where cells heavily overlap (e.g., thick tissue slices, some fluorescence images) or foreground/background distinction is not clear (TEM images, brightfield microscopy etc.), adjacency might be unclear, leading to errors in differentiation. Competing models like Cellpose can explicitly model overlapping cells using shape priors, which this method does not. The greedy four-color algorithm assumes that cells behave like planar regions. In reality, cells can have irregular or elongated shapes, making adjacency more complex. Some image applications require segmenting cell bodies and extension (e.g. hepatocytes in cytotoxic assays). The method assumes compact cell regions.

**Other Comments Or Suggestions:**

n/a

**Other Strengths And Weaknesses:**

Strength:
By proving that any cell segmentation map can be represented as a planar graph where adjacent cells can be assigned different colors using at most four labels, the authors introduce a greedy four-coloring algorithm that enforces instance separation without requiring explicit instance segmentation masks.

Weakness:
The approach makes instance differentiation emerge from the color constraints, but it’s not entirely clear how well this works for very crowded or irregularly shaped cells. As the method has no ways of incorporating expected size highly overlapping irregular shapes cell populations generate spurious cell instances (see Figure 9 in supplementary)

**Questions For Authors:**

Algorithm 1 starts with a cell graph where V is the set of cells. This algorithm does not discuss how V is obtained from the foreground. Does the algorithm assumes that each connected component in the foreground will be corresponding to a cell instance? If that is the case then why cell coloring is so important as identifying adjacent cells would be less relevant.

**Relation To Broader Scientific Literature:**

The paper falls into broader category of cell instance segmentation. The paper challenges the need for explicit instance detection by showing that cells can be differentiated purely via semantic segmentation with a constrained label space (four colors).

No prior work is cited on using four-color encoding for instance segmentation, leaving its connection to past literature vague. Similar theorems were previously used in image processing and graph coloring etc.

**Theoretical Claims:**

- Any planar graph can be colored with at most four colors such that no two adjacent regions share the same color. The authors provide a proof sketch that shows how cell segmentation map can be transformed to a planar graph, where each cell is a node and each cell boundary defines an edge.

- The four-color encoding is non-unique. This ambiguity can cause inconsistencies during training. A detailed proof is provided in the supplementary material (not checked) which shows that a mapping function that  transforms the predicted results to a four-color encoding does exist.

---

> ### Author Rebuttal · Authors · 2025-03-27
>
> We sincerely appreciate the reviewer’s thoughtful and thorough evaluation of our work and your recognition of its methodological and theoretical contributions. To address your concerns regarding real-world applicability, baseline completeness, and so on, **we summarize these issues into five points and provide detailed responses as follows**. For ease of reference, we kindly refer you to our response file: [Supp](https://drive.google.com/file/d/1dERoIrcnDklZhwcJDs5zXBBAFn7-I5u3/view?usp=sharing).
>
> **1. Generalization in Complex Scenarios**
>
> You are right that cell segmentation models should adapt to real-world challenges. To validate the generalization of FCIS, we conducted additional evaluations on complex cell segmentation datasets (recommended in CellPose 3.0) and natural scenes as shown in **Supp.RG-Tab.1**, which covers: **(1) Irregularly shaped cells** (MP6843); **(2) Low contrast, densely packed cells** (Yeaz-BF, Yeaz-PC), and **(3) Natural scenes with dense objects** (PerSense).
>
> The quantitative results in **Supp.RG-Tab.1** and qualitative visualizations in **Fig.2–6** show that our method performs robustly across these datasets, maintaining competitive AJI and DQ scores. For instance, on the Yeaz-BF and Yeaz-PC datasets, we achieve AJI scores of 0.834 and 0.822, respectively, highlighting the model's superior ability to fit complex scenarios.
>
>  **2. Motivation and Practical Relevance of FCIS**
>
> The initial motivation behind our model may differ somewhat from that of CellPose. **As a general-purpose cell segmentation model, CellPose has demonstrated remarkable segmentation performance and proven its strong generalization capability.** However, when initially employing cell segmentation approach specifically for pathology, we found some new challenges that need to be solved.
>
> - **Inference efficiency is critical**: As shown in **Supp.R3-Fig.8**, WSIs comprise tens of thousands of patches. A classical pathology segmentation model, HoVerNet, heavily relies on post-processing and causes computationally expensive (**4.59s** per patch). In contrast, our method achieves a significantly faster inference speed (**0.29s per patch**) than HoVerNet.
>
> - **Annotation scalability**: Expert-annotated cell are costly to scale, hence some **semi-supervised (SS)** or **domain-adaptive (DA)** training strategies are necessary to simplify the manual annotation. However, present SS [MMT-PSM, MICCAI] or DA [PDAM, CVPR] frameworks are very complicated shown in **Fig.9-10** due to the multi-branch or detection frameworks. Therefore, our method reformulates instance segmentation as a semantic task, which can provide new training paradigm and makes instance segmentation elegant, similar to semantic segmentation.
>
> **3. Inclusion of CellPose as a Baseline**
>
> We sincerely thank you for pointing out the omission of CellPose. In response,
> - We have included **CellPose** in our experiments (**Supp.R3-Tab.3**). Results show that while CellPose performs well across many scenarios, significantly beyond previous cell segmentation model, such as DCAN and NucleiSegNet.
> - In the revised manuscript, we will explicitly cite the **latest version of CellPose (2025, Nature Methods)** and expand the Introduction to include a more detailed discussion of its contributions.
>
> **4. Clarification on Handling Overlaps and Graph Construction**
>
> - **Overlapping Instances**: In datasets with single-instance-per-pixel annotations (e.g., most cell datasets), our four-color encoding is directly applicable. For datasets where pixels belong to multiple instances (e.g., cervical smear images), a multi-channel prediction mechanism may be more appropriate. This is a promising direction for future work.
>
> - **Node Definition in Algorithm 1**: In our method, **the cell graph is built from ground truth instance annotations**, where each connected region (cell) is assigned a unique ID. The vertex set \( V \) consists of these labeled cells, and adjacency is determined via edge contact. In scenarios where only binary masks are available, morphological processing (e.g., erosion) can be applied to approximate instance separation before applying our encoding scheme.
>
>  **5. Literature Citation**
>
> To our knowledge, **our work is the first to apply four-color encoding to cell instance segmentation in deep learning**. While previous image processing works may use coloring heuristics, our method introduces:
>
> - A greedy coloring algorithm with theoretical guarantees
> - An encoding transformation strategy that addresses non-uniqueness
> - A training pipeline that converts instance segmentation into semantic prediction
>
> Hence, We will enrich the Related Work section to include broader references from graph theory for segmentation.
>
> Once again, we thank you for your valuable feedback and for encouraging us to improve the completeness and clarity of our work. We hope these clarifications and additional results have addressed your concerns, and we welcome any further questions.

---

> > ### Comment · Reviewer_RU3q · 2025-04-03
> >
> > Thank you for the detailed rebuttal. I appreciate the additional experiments on challenging datasets, the inclusion of Cellpose in the benchmarks, and the clarification on graph construction and inference speed advantages. These additions strengthen the paper and address some of my initial concerns.
> >
> > Some limitations remain, particularly regarding overlapping cells and reliance on connected components or preprocessing for vertex construction, which may affect applicability in certain settings. Nonetheless, the core idea is novel, the formulation is well-motivated, and the empirical validation is now more complete. I will adjust my score to reflect these improvements.

---

> > > ### Author Response · Authors · 2025-04-04
> > >
> > > Thank you for the time and effort you dedicated to improving our work during the review process. We also sincerely appreciate your recognition of our responses during the rebuttal stage. Once again, thank you for your continued support and valuable contributions!

---

### Official Review · Reviewer_aw3n · 2025-03-13

**Overall Recommendation:** 3

**Summary:**

This paper proposes an asymptotic training architecture for cell instance segmentation based on the four-color theorem. Unlike traditional multi-class, multi-channel segmentation methods, the proposed approach follows a step-by-step process: first distinguishing foreground from background, then classifying instances within the foreground region. To enhance training stability, the method imposes orthogonal constraints on adjacent cells to address class imbalance and introduces an encoding transformation technique that maps outputs to a minimum color representation, ensuring consistent segmentation.

**Claims And Evidence:**

None

**Essential References Not Discussed:**

None

**Experimental Designs Or Analyses:**

It sounds. But I hope there is additional comparison as described in the second weakness that I wrote in the weakness section.

**Methods And Evaluation Criteria:**

Sounds good

**Other Comments Or Suggestions:**

See the strength and weakness part.

**Other Strengths And Weaknesses:**

Strengths:
The proposed method effectively applies the four-color theorem to cell instance segmentation, introducing a novel approach to the problem.
The authors provide an extensive ablation study across multiple datasets, demonstrating the robustness of their method.
The methodology is straightforward and easy to follow, making it accessible to readers.

Weaknesses & Areas for Improvement:
1. Introduction & Justification of Claims
The paper argues that distance mapping models significantly increase computational overhead and model complexity. However, there exist methods with fewer parameters or requiring fewer FLOPs. The authors should provide a clearer comparison to substantiate this claim.

2. Comparison with Competitive Methods
The authors should compare their approach with the recent methods in [1]. Since the proposed method based on the four-color theorem is not inherently limited to cell instance segmentation, it would be beneficial to demonstrate its effectiveness on additional segmentation datasets.
If the proposed method cannot be validated on diverse datasets, this work may be more suitable for a specialized medical imaging conference such as MICCAI.

3. Presentation of Complexity and Performance Metrics
Currently, the model complexity details are presented separately in Table 1, while performance metrics are distributed across Tables 2–4. Combining these into a single table would facilitate an easier comparison of performance and computational efficiency.

4. Equation Clarity and Notation Issues

- Equation (5) & (6): Does the index 𝑖 represent the image index {1,…,𝑁}?

- Equation (5): It appears that 𝑏 represents the background and 𝑓 represents the foreground. However, the notation for 𝑓 is reused in
- Equations (9) and (10), leading to potential confusion.

- Equation (5): Is all 𝑌^𝑖[:,0] assigned to 𝑌^𝑏? Please clarify.

- Equation (7): How does 𝑖 appear in 𝑌^{𝑏,𝑖} when there is no 𝑖 in the input?

- Equations (9) & (10): There is no explanation of how the feature 𝑓 is obtained. This should be explicitly described.

- Line 143: Given that 𝑖=1,…,𝑁 does this imply that 𝑖 and 𝑗 both range over {1,…,𝑁}? If so, how is 𝛽 again of size 𝑁? Additionally, the range of index 𝑗 should be explicitly defined.

- Line 306: The function 𝑓 mapping 𝑃 to 𝐶 is introduced, but it shares the same notation as a previously defined feature. Consider renaming to avoid ambiguity.

5. Minor Issues & Typographical Errors

- Line 321: "Where" → "where"

- Line 300: "n represents" → "where 𝑛 represents"

- Equation (15): "L_cls." → "L_cls"

**Overall Assessment**

The proposed method presents a novel adaptation of the four-color theorem for cell instance segmentation and achieves significant performance improvements. However, there are multiple issues related to equation clarity, notation consistency, and typographical errors. Furthermore, the lack of validation on diverse datasets and absence of comparisons with recent works such as [1] raise concerns about the generalizability and competitiveness of the approach.

Given these limitations, I am inclined to take a negative stance on this submission.

Reference:
[1] Chen, Zhen, et al. "Un-SAM: Universal Prompt-Free Segmentation for Generalized Nuclei Images." arXiv preprint arXiv:2402.16663 (2024).

**Questions For Authors:**

See the strength and weakness part.

**Relation To Broader Scientific Literature:**

This work can have a broader effect on the medical domain, but I am not sure in the machine learning field.

**Theoretical Claims:**

I did not find theoretical claims or analysis in this paper.

---

> ### Author Rebuttal · Authors · 2025-03-30
>
> We sincerely thank you for your valuable feedback and the time dedicated to reviewing our manuscript. To better convey our motivations and address your concerns, **we have organized our response into five key areas:** (1) theoretical contributions, (2) experimental analysis, (3) comparison with recent literature, (4) presentation and minor issues, and (5) other clarifications. For ease of reference, we kindly refer you to our response file: [Supp](https://drive.google.com/file/d/1dERoIrcnDklZhwcJDs5zXBBAFn7-I5u3/view?usp=sharing).
>
>  **1. Theoretical Contributions**
>
> Our paper introduces a novel instance segmentation framework based on the four-color theorem, which reformulates the traditionally complex instance segmentation problem as a four-class semantic segmentation task. This simplification leads to more efficient optimization and significantly faster inference than conventional instance-aware methods.
>
> In the submitted manuscript, we present two key theoretical contributions:
> - **Global Optimality Theory (Line 210)**: We demonstrate that the greedy algorithm for four-color encoding yields globally optimal results under the constraint of instance non-adjacency.
> - **Compatibility Theory (Line 291)**: We prove that any valid four-color encoding can be transformed into a canonical form, ensuring training stability through an encoding transformation mechanism.
>
> Meantime, the full theoretical derivations and proofs are also included in the supplementary material.
>
> In addition, **we believe our approach can potentially reshape traditional thinking in instance segmentation**. Drawing a parallel to the U-Net architecture, initially proposed for cell tracking and later widely adopted in general computer vision—we envision that our FCIS framework, backed by solid theoretical grounding and practical efficiency, can also generalize to a wide range of high-density instance segmentation tasks beyond the biomedical domain.
>
>  **2. Experimental Validation and Broader Applicability**
>
> We fully agree with your suggestion to demonstrate the broader applicability of our method. To that end, we conduct additional experiments on **three challenging cell segmentation datasets** and one **natural scene dataset**. These datasets include various challenging scenarios—irregular shapes, low contrast, and densely packed objects as illustrated in **Supp.RG-Fig.1**.
>
> From the quantitative results in **Supp.RG-Tab.1** and visualizations results in **Supp.RG-Figs.2–6**, we can see our FCIS consistently delivers strong and competitive performance across all scenarios. For instance, on the **Yeaz-BF** and **Yeaz-PC** datasets, we achieve AJI scores of **0.834** and **0.822**, respectively, highlighting the model's superior ability to differentiate instances. Moreover, on the **PerSense** dataset, our method outperforms the second-best approach by more than **2%**, clearly demonstrating its robustness and generalization capabilities beyond the medical imaging domain.
>
> **3. Comparison with Recent Methods (Un-SAM)**
>
> Thank you for highlighting the recent work **Un-SAM**. Following your suggestion, we **included Un-SAM as a baseline** in our supplementary experiments shown in **Supp.R2-Tab.2**. The results show that Un-SAM performs competitively across several datasets and significantly beyond DCAN and DoNet at least 2\% in Dice metrics. Besides, we also update the Un-SAM to our manuscript and added proper citations to acknowledge Un-SAM as a significant contribution.
>
>  **4. Presentation and Minor Issues**
>
> We appreciate your thorough reading regarding typographical errors. In response, **we carefully revised the manuscript** as follows:
>
> - Equation Clarity: All relevant equations have been reviewed and revised for clarity.
> - Index Definitions: The roles and ranges of index variables such as \(i\) and \(j\) have been clearly defined to remove ambiguity.
> - Grammar and Style: We corrected all minor grammatical and formatting issues to improve overall readability.
>
> Additionally, we restructuct the performance and complexity tables of the manuscript to enhance readability and facilitate comparisons, which also can be seen from **Supp.R2-Tab.2**.
>
> **5. Clarification on FLOPs vs. FLOPS**
>
> To convenience you understand the use of our FLOPs, we explain it as follows:
> - **FLOPs** (floating-point operations) measure the total number of computations required by a model and are used to evaluate algorithmic complexity.
> - **FLOPS** (floating-point operations per second) refer to the computational throughput of the hardware.
>
> In our paper, we specifically use **FLOPs** as a metric for comparing model inference efficiency. Hence, **a lower FLOPs value indicates reduced computational cost and improved efficiency**.
>
> Once again, we sincerely appreciate your detailed feedback. We hope the revised results can thoroughly address your concerns about our FCIS. If you have any further questions, we would be more than happy to provide additional clarifications.

---

> > ### Comment · Reviewer_aw3n · 2025-04-03
> >
> > Thank you for rebuttal. I carefully read the author’s rebuttal. I realized that I missed the results reported in the supplementary material. I increased my initial rating to weak accept.

---

> > > ### Author Response · Authors · 2025-04-04
> > >
> > > We sincerely appreciate your efforts in helping improve our work during the review process, as well as your recognition of our responses during the rebuttal stage. Thank you again for your kind support and valuable contribution!

---

### Official Review · Reviewer_JTJc · 2025-03-17

**Overall Recommendation:** 4

**Summary:**

This paper proposes an innovative cell instance segmentation method based on the four-color theorem, aiming to simplify the instance differentiation process. By conceptualizing cells as "countries" and tissues as "oceans," the authors introduce a four-color encoding scheme to ensure adjacent cell instances receive distinct labels, reformulating instance segmentation as a four-class semantic segmentation problem. To address training instability caused by the non-uniqueness of the four-color encoding, the authors design an asymptotic training strategy and encoding transformation method. Experimental results on multiple datasets demonstrate the method's advantages in segmentation performance and computational complexity.

**Claims And Evidence:**

The claims made in the paper are supported by clear experimental evidence, particularly from the results on various benchmark datasets. However, the issue of non-uniqueness in four-color encoding leads to instability during training, and the paper does not thoroughly explore how this issue impacts the model's performance. While the method shows great promise on several datasets, more experimental evidence is needed to prove its robustness under more complex and diverse cellular distributions.

**Essential References Not Discussed:**

The paper overlooks some essential recent works in the field of cell instance segmentation, especially some Transformer-based methods. Discussing these approaches would provide a more comprehensive understanding of the context for the proposed method and suggest possible directions for future improvements.

**Experimental Designs Or Analyses:**

The experimental design is solid, covering multiple datasets and using ablation studies to validate the components of the method. However, the paper does not sufficiently investigate how the non-uniqueness of the encoding affects model training in more complex or densely packed cell distributions. The authors could also further demonstrate the generalization of the method to other types of cellular datasets.

**Methods And Evaluation Criteria:**

The proposed four-color encoding method and evaluation criteria are reasonable and appropriate for the cell instance segmentation task. Through tests on multiple datasets, the paper demonstrates its method’s advantages in both performance and computational complexity. However, the experiments lack a thorough analysis of how the non-uniqueness of the four-color encoding impacts model training in more complex cellular distributions, which should be further explored.

**Other Comments Or Suggestions:**

1) The authors could further optimize the training process, especially addressing the instability caused by the non-uniqueness of four-color encoding. More experiments on its robustness in complex cell images would be valuable.
2) In future work, it would be beneficial for the authors to apply the method to more complex datasets and real-world biomedical application scenarios to further validate its broad applicability.

**Other Strengths And Weaknesses:**

The paper’s innovation is strong, and the proposed four-color theorem-based method offers practical value, especially in reducing computational complexity. The results show that the method can maintain good segmentation performance while reducing model complexity. However, the non-uniqueness of the four-color encoding remains a challenge, particularly in terms of training stability, which needs to be addressed in future work.

**Questions For Authors:**

1) Regarding the non-uniqueness of four-color encoding:
The paper mentions that the non-uniqueness of four-color encoding may cause training instability. Could you provide more detailed experimental results on how this issue affects model convergence, especially in complex or dense cell distributions?
2) Adaptability to complex cell distributions:
Given the strong performance on simpler datasets, could you discuss how the method performs on datasets with more complex or highly overlapping cell distributions? Are there any improvements or adaptations to handle such scenarios?
3) Future improvements to the method:
Are there any planned future improvements to handle the limitations of four-color encoding, particularly in cases where cell boundaries are irregular or cells are densely packed? How do you plan to address such edge cases?

**Relation To Broader Scientific Literature:**

The paper clearly compares its approach with existing methods in cell instance segmentation, including detection-based, contour-based, and distance mapping approaches. While these comparisons are adequate, there is room for further discussion on newer related works, particularly Transformer-based segmentation methods, which might perform better on complex datasets.

**Theoretical Claims:**

The application of the four-color theorem in cell instance segmentation is innovative. However, the discussion on the non-uniqueness of the four-color encoding and its impact on training stability is insufficient. While the asymptotic training strategy is reasonable, more in-depth theoretical explanations are needed, especially on how the strategy performs on complex datasets.

---

> ### Author Rebuttal · Authors · 2025-03-26
>
> We sincerely appreciate your thoughtful feedback, recognition of the novelty of our proposed FCIS based on the four-color theorem, and positive feedback regarding our experimental design and theoretical contributions. According to your mentioned several issues in the *Weaknesses*, *Comments*, and *Questions* sections, **we summarize these into the following four points and provide detailed responses one by one**. For ease of reference, we kindly refer you to our response file: [Supp](https://drive.google.com/file/d/1dERoIrcnDklZhwcJDs5zXBBAFn7-I5u3/view?usp=sharing).
>
>  **1. How Non-Uniqueness of Encoding Affect Training Convergence**
>
> In medical images, cells often form similar spatial patterns, such as chain-like structures. However, due to the non-uniqueness of the four-color (FC) encoding, structurally similar regions may be assigned inconsistent color configurations across different training batches.
>
> As illustrated in **Supp.R1 Fig.6**, a specific arrangement of adjacent cells may be encoded as “red-green” in batch *n*. In contrast, in batch *n+1*, it may appear as “red-blue.” This variation causes the model to receive conflicting signals during training: it may first learn that red borders green, then later that red borders blue. These inconsistencies disrupt the learning process, leading to unstable optimization and even fragmented segmentations during inference.
>
> To resolve this, we introduce an encoding transformation mechanism that includes a " buffer region" in the prediction space. This mechanism maps all variant encodings to a canonical and stable representation, i.e., the greedy FC encoding. As shown in **Supp.R1 Fig.7(a)**, this transformation significantly improves training convergence. Additionally, we provide theoretical support in the main text that the FC encoding is compatible with any other valid encoding strategy, ensuring the stability and correctness of this transformation process.
>
> To further support this claim, we provide visual comparisons in **Supp.R1 Fig.7(b)**. The segmentation results clearly demonstrate that the encoding transformation mechanism eliminates fragmented predictions, which confirms its effectiveness in enhancing model robustness.
>
> **2. Extending the Method to More Complex Scenarios**
>
> We sincerely appreciate your suggestion to evaluate our method under more complex scenarios. In response, we conduct extensive experiments on four additional datasets that cover challenging conditions: **irregular shapes, low-contrast boundaries, and densely packed regions**, along with a **natural scene dataset, PerSense**, containing many tightly clustered objects as shown in **Supp.RG-Fig.1**.
>
> The quantitative comparisons are shown in **Supp.RG-Tab.1**, and the corresponding visualization results are presented in **Supp.RG-Figs.2–6**. From the table, our FCIS achieves the best performance in DQ and PQ metrics across nearly all datasets. This demonstrate that our method **generalizes well to diverse and challenging scenarios**, maintaining high segmentation accuracy. In addition, the excellent segmentation performance from visualization results can further support the conclusions.
>
> **3. Future Directions for Improvement**
>
> As you emphasized, evaluating robustness under more complex scenarios is crucial. In response, our new experiments on four diverse datasets further demonstrate the reliability of our approach. Beyond performance, our work is also driven by a practical motivation: reducing **inference time complexity**.
>
> As discussed in our main text and **Supp.R3-Fig.8**, while distance-based methods often perform well, they require complex post-processing, making them computationally expensive. This becomes a bottleneck for applications such as whole-slide image (WSI)-level cell segmentation, where a single WSI may contain tens of thousands of patches. Our method eliminates this bottleneck, achieving inference speeds comparable to semantic segmentation—**0.29s per patch**, versus **0.94s for CellPose** and **4.59s for HoverNet**. Therefore, in the future, we will expand the method to pathological field to accelerate the analysis efficiency.
>
> Besides, we also note that high-density instance segmentation is a common challenge in domains beyond biomedicine, e.g., natural scene parsing and remote sensing. However, existing methods in these fields often rely on heavy Mask R-CNN  architectures. In contrast, our work offers a **lightweight and scalable alternative**, which will be our explored direction.
>
> **4. Other Minor Questions**
>
> We thank you for highlighting recent Transformer-based segmentation methods. We will incorporate these works into the related work section of our revised manuscript. Additionally, we will further explore the integration of Transformer modules into our framework to further enhance model performance.
>
> We hope that our responses and the newly added experiments have addressed your concerns. Once again, we sincerely thank you for your thoughtful feedback.

---

> > ### Comment · Reviewer_JTJc · 2025-04-06
> >
> > I have read the author response and agree to increase my score.

---

> > > ### Author Response · Authors · 2025-04-07
> > >
> > > Thank you for acknowledging our rebuttal and for your positive assessment of our manuscript. We sincerely appreciate your careful review of our work and the constructive comments you provided.

---

### Decision · Program_Chairs · 2025-05-01

**Decision:**

Accept (poster)

**Comment:**

This paper presents a novel application of the four-color theorem to cell instance segmentation by reformulating the problem as a constrained four-class semantic segmentation task.  The method simplifies instance differentiation, improves efficiency, and is supported by strong theoretical grounding and experimental validation.

All reviewers seem to agree that the approach is innovative and has value to the community. Some concerns were raised about generalisability to more complex or overlapping cell structures, clarity of equations, and missing baselines like Cellpose. However, the authors addressed these in the rebuttal with additional experiments and clarifications, leading most of the reviewers to raise their scores.

The idea is original, the paper is technically sound, and the method has practical value for biomedical image analysis. It makes a meaningful contribution that fits well within the ICML program.